# Absolute thermometry of human brown adipose tissue by magnetic resonance with laser polarized 129Xe

Le Zhang[1,2,4,7], Michael Antonacci [1,2,5,7], Alex Burant[1,2,6,7], Andrew McCallister[1,2], Michele Kelley[1,2], Nicholas Bryden[1,2], Christian McHugh [1,2], Sebastian Atalla[1,2], Leah Holmes[1,2], Laurence Katz[3] & Rosa Tamara Branca [1,2✉]

## Abstract

**Background** Absolute temperature measurements of tissues inside the human body are difficult to perform non-invasively. Yet, for brown adipose tissue (BAT), these measurements would enable direct monitoring of its thermogenic activity and its association with metabolic health.

**Methods** Here, we report direct measurement of absolute BAT temperature in humans during cold exposure by magnetic resonance (MR) with laser polarized xenon gas. This methodology, which leverages on the sensitivity of the chemical shift of the 129Xe isotope to temperature-induced changes in fat density, is first calibrated in vitro and then tested in vivo in rodents. Finally, it is used in humans along with positron emission tomography (PET) scans with fluorine-18-fluorodeoxyglucose to detect BAT thermogenic activity during cold exposure.

**Results** Absolute temperature measurements, obtained in rodents with an experimental error of 0.5 °C, show only a median deviation of 0.12 °C against temperature measurements made using a pre-calibrated optical temperature probe. In humans, enhanced uptake of 129Xe in BAT during cold exposure leads to background-free detection of this tissue by MR. Global measurements of supraclavicular BAT temperature, made over the course of four seconds and with an experimental error ranging from a minimum of 0.4 °C to more than 2 °C, in case of poor shimming, reveal an average BAT temperature of 38.8° ± 0.8 °C, significantly higher ($p < 0.02$ two-sided $t$ test) than 37.7 °C. Hot BAT is also detected in participants with a PET scan negative for BAT.

**Conclusions** Non-invasive, radiation-free measurements of BAT temperature by MRI with hyperpolarized 129Xe may enable longitudinal monitoring of human BAT activity under various stimulatory conditions.

### Plain language summary

Brown adipose tissue (BAT) is a fat tissue specialized in heat production and considered a potential target for the treatment of obesity and diabetes. Detection of this tissue and its metabolic activity in adult humans is challenging as this tissue is often mixed with white fat, which makes up most of the fat in adult humans. Here we demonstrate that magnetic resonance imaging with laser-polarized xenon gas, a medical imaging technique used to assess lung ventilation function, can detect the presence of this tissue in humans and measure its temperature. These temperature measurements, which show that brown fat becomes significantly hotter than 37 °C when humans are exposed to cold, may be useful in future studies to assess the effects of drugs that aim to target BAT's heat-generating activity to regulate blood sugar level.

[1] Department of Physics and Astronomy, University of North Carolina at Chapel Hill, 27599 Chapel Hill, NC, USA. [2] Biomedical Research Imaging Center, University of North Carolina at Chapel Hill, 27599 Chapel Hill, NC, USA. [3] Department of Emergency Medicine, University of North Carolina at Chapel Hill, 27599 Chapel Hill, NC, USA. [4] Present address: Small Animal Imaging Laboratory, Department of Cancer Physiology, H. Lee Moffitt Cancer Center and Research Institute, Tampa, FL 33612, USA. [5] Present address: Department of Physics, Saint Vincent College, 300 Fraser Purchase Rd., Latrobe, PA 15650, USA. [6] Present address: Department of Physics, University of Arizona, 1118 E Fourth StreetPO Box 210081 Tucson, AZ 85721, USA. [7] These authors contributed equally: Le Zhang, Michael Antonacci, Alex Burant. ✉email: rtbranca@unc.edu

Brown adipose tissue is a type of fat that most mammals have developed over the years to maintain core body temperature when they are exposed to mild cold[1]. During cold exposure, the binding of locally released catecholamines to adrenergic receptors increases intracellular cyclic adenosine monophosphate and stimulates lipolysis, while the uncoupling protein 1 allows protons to move down their electrochemical gradient, bypassing adenosine triphosphate synthase and dissipating energy as heat. This process, known as non-shivering thermogenesis, is supported by a specific increase in BAT perfusion, which is meant to meet the increased oxygen demand during heat production and to rapidly redistribute the heat produced to the rest of the body[2,3]. In rodents, BAT has been shown to regulate not only temperature, but also energy expenditure, susceptibility to weight gain, and glucose homeostasis[4–8]. A similar role is expected in adult humans, where activation of BAT by cold exposure has been shown to lead to an increase in resting energy expenditure[9,10] and to a reduction of hyperglycemia[11] and hyperlipidemia[12]. For these reasons, BAT is now considered a new target for the treatment of obesity and type-2 diabetes[13–15].

BAT temperature measurements are poised to provide the most direct way to assess BAT thermogenic function[16,17]. Yet, direct temperature measurements of BAT in humans are notoriously challenging. Optical techniques have a limited penetration depth and measurements of supraclavicular BAT temperature in humans during cold exposure have been confounded by vasoconstriction and varying subcutaneous fat thickness[18].

Magnetic resonance (MR) thermometry is commonly used for in vivo non-invasive temperature measurements in tissues. The most widely used MR temperature probe is the water resonance frequency, which shifts linearly with temperature by $-0.01$ ppm/$^{\circ}$C[19]. However, the NMR frequency of nuclei in tissues is primarily determined by the strength of the external magnetic field and by the local field, which results from the perturbation of the external magnetic field by the sample. These perturbations, which originate from differences in magnetic susceptibility between air and tissues and between different tissues, depend on the tissue's structure and composition, as well as on its orientation with respect to the magnetic field. The effects of these distortions are only partially rectified by shim gradients and can be removed if a temperature-insensitive resonance frequency can be found in the same tissue compartment that "feels" the same field distortions. For example, the $^{1}$H resonance of the N-acetyl-aspartate methyl group has been successfully used in the brain as a temperature-independent internal reference frequency for the temperature-sensitive water protons to enable the extraction of absolute temperature information[20,21]. In other parts of the body, a similar strategy has proven to be unsuccessful because the methylene resonance frequency of lipid protons (the CH$_2$ peak), which is the second largest resonance seen in in vivo $^{1}$H NMR spectra, originates from a different tissue/cell compartment than water $^{1}$H, and is thereby subjected to a different local field distortion[22,23]. Additionally, the magnetic susceptibility of fat has a temperature dependence which ranges from 0.004 to 0.008 ppm/$^{\circ}$C[24], which makes these field distortions temperature-sensitive and affects the accuracy of both absolute and relative temperature measurements made by using the water resonance frequency[22,25–27]. For these reasons, in fatty tissues, MR thermometry methods based on the temperature-induced changes of the longitudinal relaxation of water and lipid protons' magnetization are generally preferred. However, because the longitudinal relaxation depends both on the molecular composition of the tissue and the field strength, this method requires pre-calibration of the longitudinal relaxation and its temperature dependence in the tissue of interest at the working field strength[25,28,29].

Here we report the direct detection of human brown adipose tissue temperature by magnetic resonance with laser-polarized $^{129}$Xe (here abbreviated as HPXe). To the best of our knowledge, these are the first temperature measurements of human BAT made with laser-polarized $^{129}$Xe. For these measurements, we first enhance the nuclear-spin polarization of the MR-visible $^{129}$Xe isotope more than 25,000-fold with respect to thermal equilibrium through Spin Exchange Optical Pumping (SEOP)[30], a process whereby angular momentum from circularly polarized light is indirectly transferred to the nuclear spin of $^{129}$Xe atoms. This enhancement enables the detection of $^{129}$Xe dissolved in distal organs at the micromolar concentrations typically reached in humans after a single inhalation of the gas.

There are a few properties that make the $^{129}$Xe isotope a unique MR probe for the detection of brown fat. First, the $^{129}$Xe nuclear spin frequency is easily modified by short-range contacts with nearby atoms and molecules with which $^{129}$Xe atoms may come in close proximity in the tissue of interest[31]. This enables researchers to clearly differentiate, in the $^{129}$Xe NMR spectrum, the signal originating from xenon dissolved in fat (here abbreviated as LDX) from the signal originating from xenon dissolved in muscle or blood[3,32]. Particularly in blood, $^{129}$Xe rapidly diffuses between plasma and hemoglobin, giving rise to two distinct resonances that are about 20 ppm away from each other. Second, because xenon is an inert gas that can reach distal organs only through the vasculature, the concentration reached in each tissue is determined solely by the tissue's blood perfusion and by the xenon solubility in the tissue of interest. While the solubility of xenon in fat is twenty fold higher than in any other tissue compartment[32,33], low blood perfusion generally limits the concentration of xenon reached in fat after a single inhalation of the gas[34]. On the other hand, detectable concentrations of xenon are expected to be found in highly perfused fat tissues like brown fat, especially during stimulation of thermogenesis, when tissue blood flow increases[35–38]. Finally, the LDX chemical shift is very sensitive to the temperature-induced changes in the density of fat[39,40]. The linear temperature dependence of the chemical shift of LDX, recently measured to be $-0.2$ ppm/$^{\circ}$C[39,40], is twenty-fold higher than that of water protons and enables the direct measurement of temperature changes in fatty tissues as well as absolute temperature. The latter can be obtained by referencing the resonance frequency of xenon dissolved in the lipid compartment of BAT to that of nearby methylene protons (CH$_2$) of BAT's triglycerides to remove the effect of macro and microscopic magnetic susceptibility gradients, while retaining the temperature-sensitive information encoded in the LDX chemical shift. This dimensionless and field-independent CH$_2$-referenced LDX frequency, here abbreviated as $\omega_{rLDX}$, has already been used to directly measure BAT temperature during stimulation of non-shivering thermogenesis in rodents by catecholamines[40–42].

The translation of this methodology to humans presents several challenges. First, unlike in rodents where the HPXe gas can be provided continuously over the course of several minutes, enabling saturation of HPXe signal in distal tissues, in humans, HPXe gas is generally delivered via a single breath-hold, limiting the HPXe concentration in tissue, and thereby the achievable signal. Second, the longer blood transit time in humans is expected to lead to higher xenon depolarization, especially at clinical field strengths[43]. It is unclear whether the higher depolarization, coupled with the much smaller increase in BAT blood flow reported in humans during cold exposure[3] can still lead to a detectable lipid-dissolved $^{129}$Xe signal in BAT for its detection by MR at clinical field strengths. Third, since BAT temperature in humans has never been measured directly, it is unclear whether the accuracy and sensitivity of the hyperpolarized $^{129}$Xe MR thermometry method is sufficient to detect the much smaller

increase in tissue temperature expected in humans during cold exposure[17].

Here, to answer these questions, after calibration of the LDX temperature probe in excised human adipose tissue samples and validation of this absolute thermometry method in rodents, we directly apply this technique to humans to detect BAT and to measure its temperature during cold-induced stimulation of thermogenic activity. We find that cold exposure leads to a selective accumulation of hyperpolarized $^{129}$Xe in the lipid compartment of BAT, enabling background-free detection of this tissue by MRI, while absolute temperature measurements performed by using combined HPXe and $^1$H NMR spectroscopy reveal active BAT thermogenesis even in persons that show a PET scan negative for BAT.

## Methods

**Calibration of the lipid-dissolved xenon NMR resonance for absolute MR thermometry.** Calibration of the field independent $CH_2$-referenced LDX frequency ($\omega_{rLDX}$) used for absolute thermometry measurements was done by re-analyzing four de-identified human adipose tissue NMR datasets, previously collected in our laboratory and presented in Antonacci et al.[39]. As specified in Antonacci et al.[39], these samples had been obtained from 4 patients undergoing plastic surgery. For the NMR measurements, tissue samples stored in ice were immediately brought into the NMR suite, where they were placed in a high-pressure NMR tube (Daedalus Innovations, LLC, Aston, PA) connected to a home-built vacuum pump. To ensure that all oxygen was removed from the sample, samples were frozen in liquid nitrogen, evacuated with a rotary pump, and thawed. After three freezing and thawing cycles, the head spaces of the NMR tubes were evacuated, and the samples were exposed to about 1 atm of 86% isotopically enriched $^{129}$Xe gas. This low pressure was used to avoid changes in chemical shift due to Xe–Xe interactions[40]. Samples were then sealed and inserted into a high-resolution 500 MHz NMR spectrometer (Varian NMR Systems, Palo Alto, CA) where they were equilibrated for approximately 1 h at temperatures of approximately 25 °C, 30 °C, 35 °C, and 40 °C. These temperatures were calibrated with an accuracy of 0.1 °C at the beginning of each NMR study by using a 100% methanol temperature standard. Right before each measurement, all samples were manually shimmed using the $CH_2$ resonance. Proton spectra were acquired before and after the acquisition of each $^{129}$Xe spectra to confirm consistent conditions throughout the $^{129}$Xe acquisition. $^{129}$Xe NMR spectra were acquired with a repetition time (TR) = 26 s, spectral width (SW) = 400 ppm, 36 averages (NA), and 46° flip angle, while $^1$H NMR spectra were acquired with a TR = 6 s, SW = 12 ppm, a flip angle of 6° and 16 averages.

For the re-analysis, the center frequencies of the $CH_2$ and LDX resonances, $\omega_{CH_2}$ and $\omega_{LDX}$, respectively, were measured by using the peakfit fitting procedure using MATLAB R2017b software (MathWorks, Natick, Massachusetts, USA), while bootstrapping was used to determine the uncertainty in the peak position. The LDX resonance was then referenced to the $CH_2$ resonance to extract the dimensionless, field-independent, parameter:

$$\omega_{rLDX} = \frac{\left(\omega_{LDX} - \omega_{CH2}\frac{\gamma_{Xe}}{\gamma_H}\right)}{\omega_{CH2}\frac{\gamma_{Xe}}{\gamma_H}} \quad (1)$$

with $\frac{\gamma_{Xe}}{\gamma_H} = 3.615295022$. In the above expression, the normalization by $\omega_{CH2}\frac{\gamma_{Xe}}{\gamma_H}$ was used to obtain a field-independent parameter, while the rescaling of $\omega_{CH2}$ by $\frac{\gamma_{Xe}}{\gamma_H}$ was simply done out of convenience to obtain a similar temperature-induced frequency shift for $\omega_{rLDX}$ as that of $\omega_{LDX}$.

A linear regression of the 16 (4 values per sample) $\omega_{rLDX}(T)$ values, calculated by using expression (1), with the model:

$$T(°C) = \frac{(-\omega_{rLDX} + a)}{b} \quad (2)$$

was finally used to calculate the relation between $\omega_{rLDX}$ and tissue's absolute temperature.

**In vivo localized absolute MR thermometry measurements.** In vivo animal experiments were performed to assess the accuracy of the temperature-dependent field-independent parameter $\omega_{rLDX}$ and its robustness under in vivo conditions. All animal procedures were performed according to the ethical guidelines for animal experiments as described in the Public Health Service Policy on Humane Care and Use of Laboratory Animals, the Animal Welfare Act and Animal Welfare Regulations, and the Guide for the Care and Use of Laboratory Animals, under an animal protocol approved by the Institutional Animal Care and Use Committee (IACUC) at the University of North Carolina at Chapel Hill. The methods reported here also adhere to ARRIVE guidelines. For these experiments, 4 ob/ob female mice, 5-month-old and weighing between 65 and 75 g, were purchased from Jackson Laboratory (Bar Harbor, ME, USA) and used for the in vivo NMR spectroscopy experiments, which were all performed on a 9.4 T MRI horizontal bore small animal magnet (Bruker BioSpec 94/30, Bruker Biospin Corp., Billerica, Massachusetts, USA). Because the purpose of these experiments was to assess the accuracy of the $\omega_{rLDX}$-based thermometry method as animal body temperature and shimming conditions were varied, multiple measurements were made in the same animal at different temperatures and shimming conditions such that each animal served as its own control and no control group was included in the experimental design. To mimic the detection conditions used in humans, after inducing general anesthesia by using an intraperitoneal injection of pentobarbital (70 mg/kg dose), mice were placed with their rectum on top of a small $^{129}$Xe surface coil (m2m Imaging Corp.) with a 1 cm diameter. The mouse and the $^{129}$Xe surface coil were then placed inside an 86 mm $^1$H quadrature volume coil (Bruker Biospin Corp.). An i.p. catheter was used to provide additional anesthesia as needed. Bore temperature was controlled by a MR-compatible thermocouple and varied over the course of the entire experiment by a forced-air heating system (Small Animal Instruments). A pre-calibrated fiber optic temperature probe (Osensa Innovations Corp., Canada) was placed in the rectum of the anesthetized mice, within the sensitive region of the small $^{129}$Xe surface coil and the selected $^1$H voxel. Hyperpolarized xenon was delivered to the animal by using a mechanical ventilator, at a rate of 80 breaths per minute, with a tidal volume of 0.15 mL, and a content of 30-vol% $O_2$ and 70-vol% hyperpolarized $^{129}$Xe. For the temperature measurements, animals were equilibrated at different bore temperatures for 20–30 min before the NMR temperature measurements were made. The number of measurements performed on each animal varied between animals (from a minimum of 6 up to 20 in one animal). Anatomical images were acquired for reference, and manual shimming procedures were performed on the $CH_2$ resonance on a volume located right above the $^{129}$Xe surface coil, encompassing the rectal probe. In most cases, re-shimming was performed between measurements to ensure narrow spectral line. In all cases, localized $^1$H spectra (Volume 2 × 2 × 1.5 cm) and unlocalized $^{129}$Xe spectra were acquired back-to-back from a smaller volume using the same shimming conditions. For these terminal studies, animals were not recovered from anesthesia but euthanized by an overdose of pentobarbital (200 mg/kg) at the end of the last temperature measurement.

After the acquisition of $^1$H and $^{129}$Xe spectra, a peak fitting procedure (peakfit) using MATLAB R2017b software (Math-Works, Natick, Massachusetts, USA) was used to identify the $CH_2$ and the LDX peak position and the associated uncertainty $\sigma_{\omega_{LDX}}$ and $\sigma_{\omega_{CH2}}$ from the $^{129}$Xe and $^1$H spectra that had been collected, while blinding the animal's temperature data. The uncertainty on $\omega_{rLDX}$ was then computed using the expression:

$$\sigma_{\omega_{rLDX}} = \sqrt{\left(\frac{1}{\omega_{CH2}\frac{\gamma_{Xe}}{\gamma_H}}\right)^2 \sigma_{\omega_{rLDX}}{}^2 + \left(\frac{\omega_{LDX}}{\omega_{CH2}{}^2\frac{\gamma_{Xe}}{\gamma_H}}\right)^2 \sigma_{\omega_{CH2}}{}^2} \quad (3)$$

A paired, two-sided $t$ test was then used to assess differences between temperature measurements made by using the LDX NMR probe, and temperature measurements made by the optical probe.

**Human study protocol.** Human studies were approved by the Institutional Review Board of the University of North Carolina at Chapel Hill (IRB # 15-0749) and were conducted in compliance with the Declaration of Helsinki and under an Investigational New Drug application for BAT imaging with laser-polarized xenon gas. A total of 18 participants (10 female and 8 male) were recruited through flyers and mass emails sent to staff and students at the University of North Carolina at Chapel Hill via the University Listserv application or through the Healthy Volunteer Database used for studies conducted at the Biomedical Research Imaging Center. In accordance with the Declaration of Helsinki, all participants gave their written informed consent to participate in this study, which was powered to assess the feasibility of detecting human brown adipose tissue with hyperpolarized $^{129}$Xe MRI (contrast to noise ratio of hyperpolarized $^{129}$Xe signal in supraclavicular fat ≥5) with a 95% confidence level. All participants were aged 20–34 y.o. (with a median of 26 y.o.), had a BMI between 17.7 and 29.9 kg/m$^2$ (median of 22.9 kg/m$^2$), were non-smokers, had no known metabolic or psychological conditions, and did not take medications that are known to interfere with BAT activity.

A cartoon showing the main idea of the methodology is shown in Fig. 1 while a flowchart of the human study design is shown in Fig. 2. All participants underwent one or more HPXe MR scans, followed by either an $^{18}$F-FDG-PET/MRI scan (10 participants) or an $^{18}$F-FDG-PET/CT (6 participants) scan, all within one week. For the HPXe MR scan, participants were neither asked to fast nor to abstain from exercise before the scan, whereas for all $^{18}$F-FDG PET studies, participants were asked to fast for at least 6 h and abstain from intense exercise for at least 24 h prior to the PET study.

**Hyperpolarized $^{129}$Xe and $^1$H MRI measurements in humans.** MR spectra and images were acquired on a 3 T MAGNETOM TRIO scanner (Siemens Healthineers, Germany—5 participants), later upgraded to a 3 T MAGNETOM PRISMA scanner (Siemens Healthineers, Germany—13 participants). After screening and consenting, participants were first instructed on how to perform the xenon inhalation and breath hold procedure. They were then brought into the MR scanner room where water-perfused cooling pads (ArcticSun 2000 Temperature Management System, Medivance, Louisville, CO) were wrapped around their thighs and torso (Supplementary Fig. 1). All participants were positioned supine on the scanner table and $^1$H images were acquired using the body coil for anatomical reference. $^1$H images were acquired by using a 2-echo DIXON sequence with a square field of view (FOV) ranging from 300 to 500 mm in size, depending on the person size, a slice thickness of 1 mm, TR = 10 ms, and echo time (TE) of 2.46 and 3.69 ms, and a flip angle of 13°. Right before the

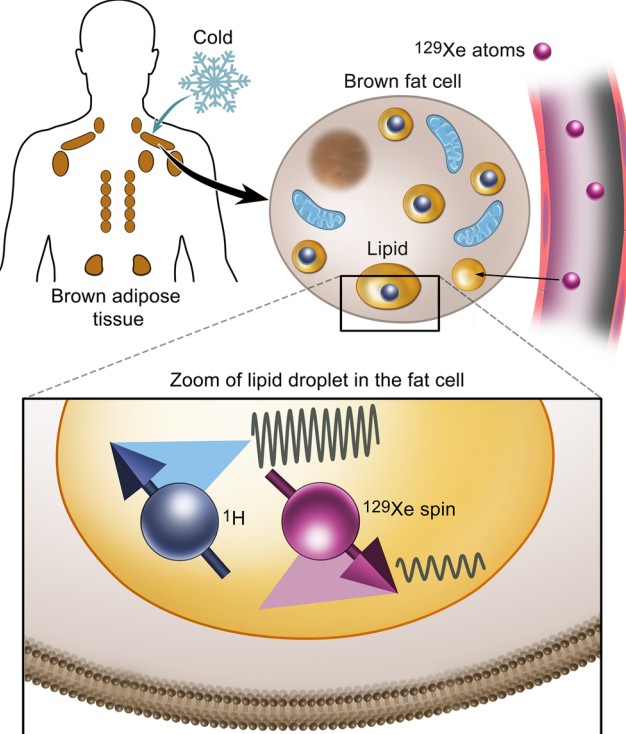

**Fig. 1 A schematic illustration of the hyperpolarized $^{129}$Xe methodology for measuring absolute temperature of BAT.** Cold exposure in adult humans leads to an increase in blood flow and transport of hyperpolarized $^{129}$Xe in BAT. The lipophilic $^{129}$Xe atoms transported by blood to BAT selectively accumulate in the lipid compartment of brown adipocytes. Combined measurements of the resonance frequency of $^{129}$Xe and lipid protons enable the removal of local magnetic field inhomogeneities and the detection of absolute tissue temperature.

acquisition of HPXe spectra or images, the participants were instructed to snap open the Tedlar bag containing hyperpolarized $^{129}$Xe gas and inhale the contents of the bag as fast as they could. To guarantee consistency, the entire inhalation procedure, which was completed within 5 seconds and was followed by a 20-second breath-hold, was coached and timed by research personnel who were observing the participant before, during, and after each HPXe gas inhalation. Before and after each HPXe inhalation, the participant's heart rate, blood pressure, and oxygen saturation level were recorded (3150 MRI Patient Monitor; Invivo Research).

HPXe spectra and images were initially acquired by using a circular, dual tuned, $^1$H/$^{129}$Xe coil (XZACT MEDICAL INC, London Ontario, Canada) with a ~8 cm inner diameter and a penetration depth ~4 cm, and later acquired by using a single-tuned $^{129}$Xe coil (either a 12.7 cm × 12.7 cm square coil with a penetration depth of ~6.5 cm, or a 3.55 cm × 3.55 cm square loop with a penetration depth of about 1.8 cm) from Clinical MR Solutions (Brookfield, WI, USA) along with the built-in body coil for $^1$H acquisitions. To minimize gas-phase contaminations, dissolved-phase xenon images were acquired during a breath hold of room air, which followed HPXe gas exhalation, with the transmit frequency centered on the lipid-dissolved $^{129}$Xe resonance and with a pulse bandwidth of 3.3 kHz. For imaging, the dissolved-phase $^{129}$Xe signal was spatially encoded by using a two-dimensional gradient-echo sequence in either axial or coronal orientation, employing a sequential phase-encoding order (nominal flip angle of 10°, a TR/TE of 25 ms /2.1 ms; a FOV ranging from 300-400 mm in size, and a matrix size (MTX)

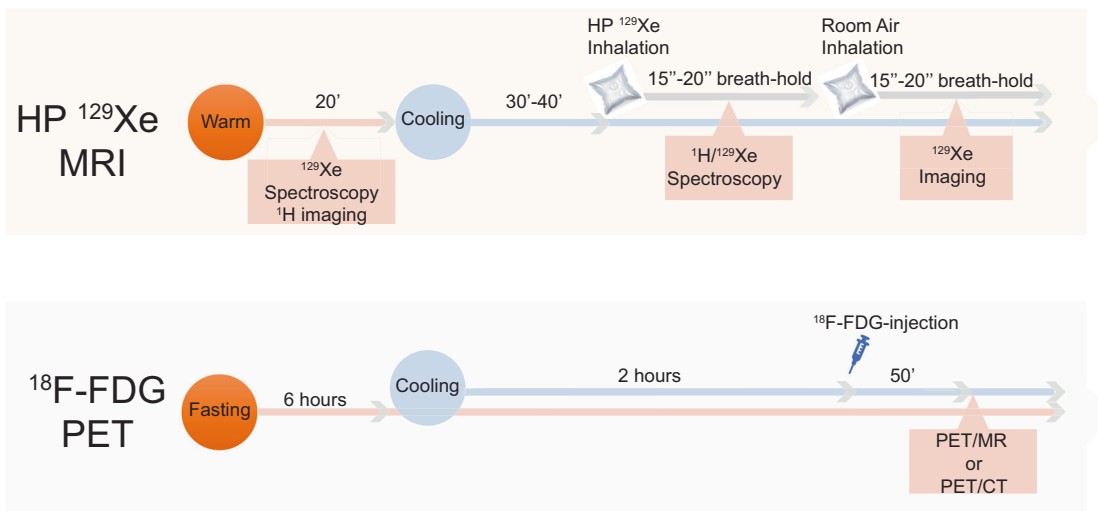

**Fig. 2 Flow chart of the human study protocol used in this study.** For the hyperpolarized xenon (HPXe) MRI procedure, participants were exposed to cold for 30–40 min before $^1$H and $^{129}$Xe spectroscopy studies were performed. In this case, water temperature was set at 18 °C and lowered until they started to shiver. When that occurred, temperature was raised until shivering subsided. Three participants were first exposed to warm condition during which a baseline $^{129}$Xe spectroscopy study was performed before cold exposure began. For the $^{18}$FDG-PET study, participants were asked to fast for at least 6 h prior to arriving at the imaging center. After changing into scrubs, they were wrapped with water perfused cooling pads (same as the ones used for the MRI study). Initial water temperature setting was 18 °C. Temperature was then lowered every 5 min until they began to shiver and then raised by 1 °C every 5 min until shivering stopped. In all cases, all participants were exposed to cold temperatures (≤18 °C) for at least 2 h before $^{18}$F-FDG injection and during the PET scan.

of 32 × 8, 64 × 16, or 64 × 32). The removal of additional gas phase signal contaminations was achieved by choosing a spectral bandwidth of 430 Hz/pixel, which pushed the gas phase signal contamination to the edges of the FOV. The total acquisition time for these images was less than 8 s and their signal-to-noise ratios (SNR) ranged from 6 to 30. Initial $^{129}$Xe images acquired on the MAGNETOM TRIO scanner were reconstructed offline by using Matlab (MathWorks) scripts, while all subsequent $^{129}$Xe images acquired on the MAGNETOM PRISMA scanner were automatically reconstructed online by the MR scanner.

**Hyperpolarized $^{129}$Xe temperature measurements in humans.** Absolute temperature measurements of BAT were performed in 15 of the 18 participants enrolled in this study. All hyperpolarized $^{129}$Xe temperature measurements were performed at least 20 min after the beginning of cold exposure, when enhanced $^{129}$Xe uptake in BAT led to a strong LDX peak. For these measurements, localized shimming was first manually performed during the participant's free breathing on the CH$_2$ peak originating from a volume encompassing the supraclavicular fat pad. Shim current configurations were recorded and manually copied for the subsequent localized $^1$H and $^{129}$Xe spectral acquisitions, which were acquired back-to-back within a 400 ms delay from each other during the first 10 s of the $^{129}$Xe breath-hold. Localized $^1$H spectra were acquired from a volume less than $4 \times 4 \times 4$ cm$^3$ located within the supraclavicular fat depot by using a point resolved spectroscopy sequence (PRESS) with a TR of 1820 ms, TE = 35 ms, NA = 2, and a vector size of 4096. Non-localized $^{129}$Xe spectra were acquired by using a rectangular 0.5 ms pulse, 2048 acquisition points and a SW of 500 ppm. Line broadening and zero filling were then applied to the raw data before a 1D fast-Fourier transform. Fitting of the $^1$H and $^{129}$Xe spectra was done in Matlab using the peakfit protocol to measure the $\omega_{LDX}$ and $\omega_{CH2}$ peak positions and their uncertainties. Equation [1] was then used to compute $\omega_{rLDX}$, while equation [2], with the $a$ and $b$ coefficients previously determined, was used to

extract absolute BAT temperature. Temperature measurement uncertainties were calculated by using error propagation analysis using the expression:

$$\sigma_T(°C) = \sqrt{\left(\frac{1}{b}\right)^2 \sigma_{\omega_{rLDX}}{}^2 + \left(\frac{1}{b}\right)^2 \sigma_a{}^2 + \left(\frac{\omega_{rLDX} - a}{b^2}\right)^2 \sigma_b{}^2} \quad (4)$$

with $\sigma_{\omega_{rLDX}}$ calculated using the uncertainty in the LDX and CH$_2$ peak positions, and $\sigma_a$ and $\sigma_b$ being the standard deviations of the previously measured $a$ and $b$ coefficients.

**$^{18}$F-FDG-PET measurements in humans.** For the $^{18}$F-FDG PET scan, the participants were instructed to refrain from strenuous activities and exercise for at least 24 h before the imaging session and to fast for a period of at least 6 h prior to arriving at the imaging center. On the day of the PET imaging, after screening and consenting, the participants were brought into one of the patient rooms located within the imaging suite. Water-perfused cooling pads (ArcticSun 2000 Temperature Management System, Medivance, Louisville, CO) were wrapped around their thighs and torso after the participants changed into scrubs. Water temperature was then lowered from its initial value of 18 °C every 5 min until the participants began to shiver. After reaching the shivering threshold, the temperature was raised by increments of 1 °C every 5 min until shivering subsided, as self-reported by the participant. In all cases, final water temperature values did not differ from those used during the MRI scan by more than 1 °C and in all cases were below 18 °C. After about 2 h of cooling, 185 MBq of $^{18}$F-FDG were injected intravenously. The participants were then cooled for an additional 50 min before being transferred to either a PET/MR scanner (Biograph mMR, Siemens Healthineers, Erlangen, Germany), or a PET/CT scanner (Biograph mCT, Siemens Healthineers, Erlangen, Germany). The choice of the scanner was mostly based on combined availability of both the scanner and the participant. PET acquisitions were performed with the participant still connected to the water-perfused cooling pads in three-dimensional mode with 6 min per

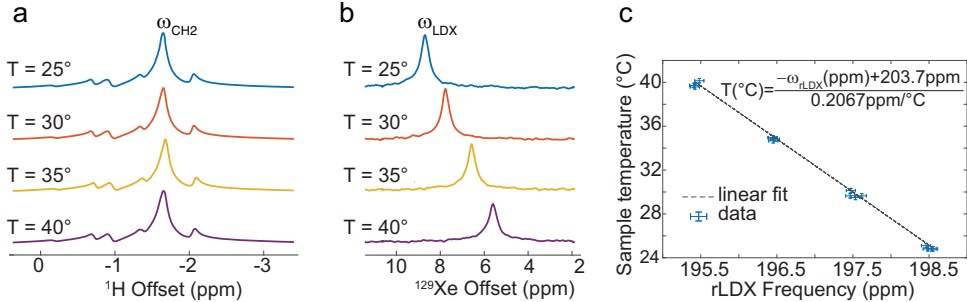

**Fig. 3 In vitro calibration of the temperature sensitive NMR $^{129}$Xe probe. a** Examples of $^1$H spectra acquired previously acquired in our lab and reported in Antonacci et al.[39] from the same human adipose tissue sample at four different temperatures that show a temperature insensitive methylene peak. **b** Corresponding $^{129}$Xe spectra acquired from the same sample at the four different temperatures. A clear up-field (i.e. toward lower frequencies) frequency shift of the lipid-dissolved xenon resonance frequency ($\omega_{LDX}$) can be observed as temperature is increased. For these measurements, the temperature-insensitive CH$_2$ frequency ($\omega_{CH2}$) is used as internal reference to remove the contribution of the magnetic field gradients from $\omega_{LDX}$ while retaining its temperature information. **c** Plot showing the CH$_2$-referenced LDX frequency as function of temperature as measured for all four human adipose tissue samples (n = 16 independent measurements acquired on the four samples and reported in Supplementary Data 1). Error bars represent measurement experimental uncertainties derived from error propagation accounting for the experimental uncertainty in the CH$_2$ and LDX peak positions. A linear regression with the model $T(°C) = \frac{-\omega_{rLDX}+a}{b}$ returned values for $a$ and $b$ with a 95% confidence bound of 203.7ppm(203.5ppm, 203.8ppm) and $0.2067\frac{ppm}{°C}(0.2016\frac{ppm}{°C}, 0.2118\frac{ppm}{°C})$ $b$, respectively. Error bars represent $\omega_{rLDX}$ measurement uncertainties.

bed position and a total of 3–4 bed positions to cover an area from the skull to the heart. PET image resolution was 4.1 mm × 2.6 mm × 3.1 mm. Anatomical data for attenuation correction were collected either simultaneously (PET/MRI) or right before (PET/CT) the acquisition of PET data. At the end of the imaging scan, the water-perfused cooling pads were removed and the participant was released. PET images were reconstructed and analyzed using MIM software (MIM Software Inc. Cleveland, OH).

**$^{129}$Xe gas polarization procedure**. For all these studies, $^{129}$Xe was polarized from an initial gas blend composed of 1% Xe (87% $^{129}$Xe), 10% N$_2$, and 89% $^4$He by continuous-flow spin exchange optical pumping followed by cryogenic extraction using a commercial xenon polarizer (Xenon polarizer, model 9800, Polarean Inc, Durham NC). $^{129}$Xe polarization was measured by using a commercial polarimeter (Polarean Inc, Durham NC) and ranged from 14–16%. For in vivo animal studies, 200 ml of HPXe was dispensed into Tedlar bags (Jensen Inert Products, Coral Springs, FL), connected to a hyperpolarized gas compatible ventilator. For human studies, 400-500 ml of polarized gas was dispensed into Tedlar bags along with N$_2$ buffer gas for a final total gas volume of about 750 ml.

**Reporting summary**. Further information on research design is available in the Nature Portfolio Reporting Summary linked to this article.

## Results
**Calibration of the CH$_2$-referenced lipid-dissolved xenon frequency for absolute MR thermometry**. Figure 3 shows $^1$H and $^{129}$Xe NMR spectra acquired from one of the human adipose tissue samples at the four different temperatures. Data shown in Fig. 3c are reported in Supplementary Data 1 file. Linear regression analysis to find the coefficients in [2] (R$^2$ = 99.8% and RMSE = 0.257) gave, with 95% confidence bounds range, values of $a = 203.7ppm(203.5ppm, 203.8ppm)$ and $b = 0.2067\frac{ppm}{°C}$ $(0.2016\frac{ppm}{°C}, 0.2118\frac{ppm}{°C})$.

**Absolute MR thermometry validation experiments in mice**. To assess the accuracy of the $^{129}$Xe-based MR thermometer method

in vivo, after the in vitro calibration studies, we used $\omega_{rLDX}$ to calculate absolute adipose tissue temperature in vivo in mice using the previously found relation between temperature and $\omega_{rLDX}$:

$$T(°C) = \frac{(-\omega_{rLDX}(ppm) + 203.7ppm)}{0.2067ppm/°C} \quad (5)$$

An example of the localized $^1$H and $^{129}$Xe spectra, consecutively acquired in anesthetized obese mice from a region encompassing the animal rectum and containing an MR-compatible fluorescence temperature probe is shown in Fig. 4. The same figure shows the comparison between temperature values measured spectroscopically using equation [5] and those measured by a pre-calibrated MR-compatible optical probe in the same region (Supplementary Data 2). The difference between the two measurements ranged from −0.3 °C to 0.4 °C, with a median difference of 0.12 °C and a standard deviation of 0.16 °C. A two-sided paired t-test, conducted to assess whether the two measurements were statistically different, revealed that the mean temperature measurement calculated by using equation [5] was different than the one provided by the optical rectal temperature probe by only 0.1 °C with a 99% confidence level.

**Cold exposure leads to selective enhancement of the lipid-dissolved HP $^{129}$Xe resonance frequency and background free detection of BAT**. Figure 5 and Supplementary Fig. 2 show some of the initial HPXe spectra and images collected in humans, along with the corresponding $^{18}$F-FDG-PET/MR images. The $^{129}$Xe NMR spectra acquired at thermoneutrality (water temperature 30 °C), 8 s from the beginning of a breath hold of 500 ml of HPXe gas, show primarily a gas-phase peak located at 0 ppm, which originates from the inhaled gas residing in the apex of the lungs and encompassing the sensitive region of the surface coil. During cold exposure, a selective increase of the lipid-dissolved $^{129}$Xe peak, located 191-192 ppm downfield (i.e. higher ppm values) from the gas phase peak, is observed in the $^{129}$Xe NMR spectra. After inhalation, this signal reaches its maximum intensity at about 16 s from the beginning of the breath hold, and slowly decays thereafter, while lasting for almost a minute (Fig. 5d). Another dissolved-phase peak, originating from an aqueous-like compartment[44], was also observed 10 ppm downfield from the LDX peak. This peak, unlike the LDX peak, was only seen when

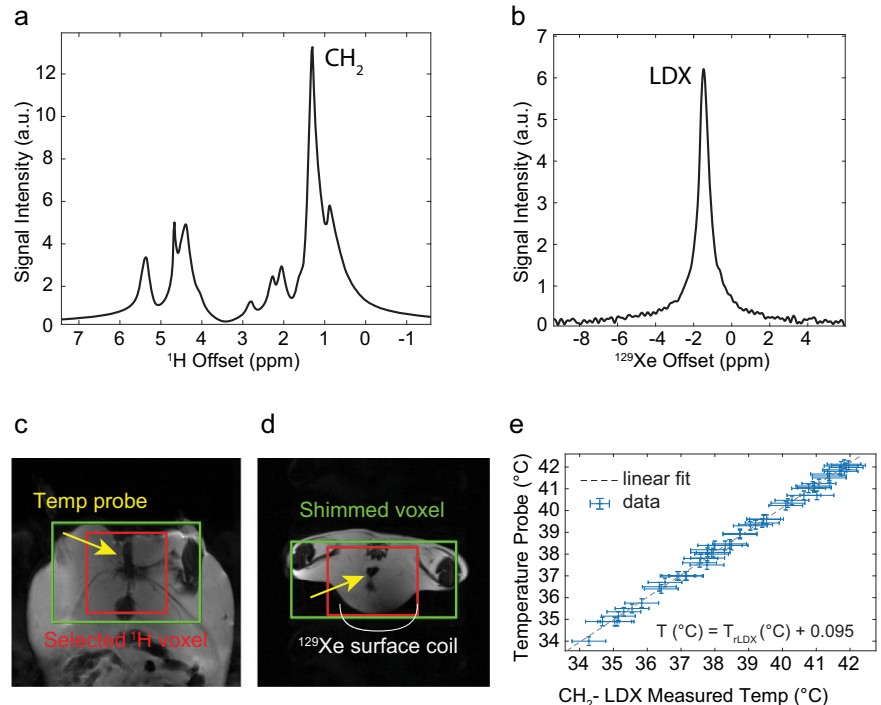

**Fig. 4 Hyperpolarized $^{129}$Xe MR thermometry in mice. a** Example of $^1$H spectrum acquired from one of the mice showing the lipid peaks along with the CH$_2$ resonance frequency line used as internal reference to measure absolute temperature in fat. **b** $^{129}$Xe spectrum acquired from the same mouse showing the temperature-sensitive lipid-dissolved xenon resonance frequency line (LDX). **c** Coronal image of one of the obese mice showing the relative location of the shimmed voxel (green box) and the voxel from which the CH$_2$ spectrum was acquired (red box). **d** Axial image of one of the obese mice showing the relative location of the shimmed voxel (green box) and the voxel from which the CH$_2$ spectrum was acquired (red box) along with the location of the $^{129}$Xe surface coil (white line) and of the optical temperature probe (yellow arrow), running through the middle of the voxel. **e** Scatter plot showing correlation between tissue temperature measured by using the CH$_2$-referenced LDX frequency $\omega_{rLDX}$ and tissue temperature measured by the optical temperature probe ($n = 54$ independent measurements acquired on the four animals and reported in Supplementary Data 2). Data collected above 40.5 °C originated from a single animal that probably entered a febrile state before being euthanized. Error bars represent measurement experimental uncertainties derived from error propagation accounting for the experimental uncertainty in the CH$_2$ and LDX peak positions, as well as on the uncertainty associated with the temperature coefficients of Eq. (2). The median difference between the two measurements was 0.12 °C and ranged from −0.32 °C to 0.14 °C, while the standard deviation was 0.16 °C.

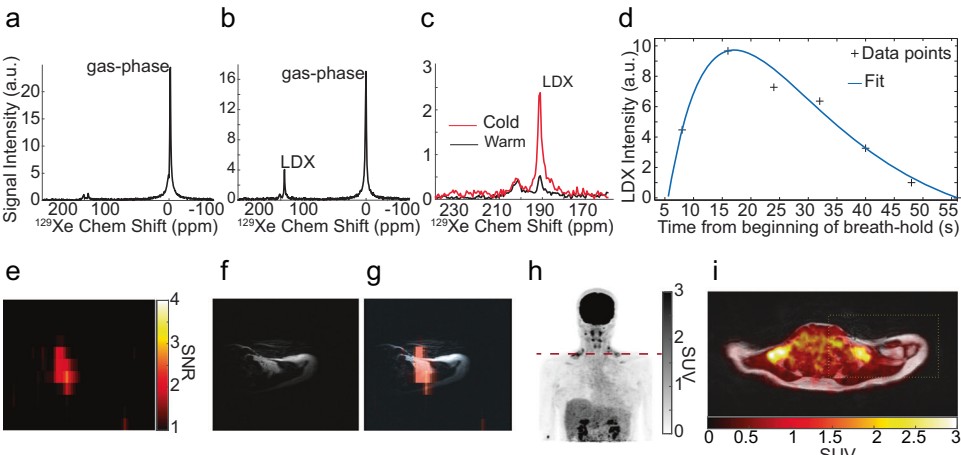

**Fig. 5 Cold exposure leads to a selective uptake of inhaled hyperpolarized xenon in supraclavicular human BAT. a** HPXe spectrum acquired at thermoneutrality, 16 s from the beginning of the breath-hold, on a 25-30 y.o. female with a BMI of 23 kg/m$^2$. **b** HPXe spectrum acquired on the same participant during cold exposure, also 16 s from the beginning of the breath-hold. **c** Zoomed-in view of spectra shown in (a) and (b) showing selective increase of the lipid-dissolved xenon (LDX) peak, normalized to the corresponding gas-phase peak, during cold exposure. **d** Plot showing the expected time-dependent signal function (see ref. [56]) fitted to the acquired LDX signal. **e** Hyperpolarized xenon (HPXe) MR image acquired on the same participant after HPXe exhalation, over the course of 16 s (field of view of 300 × 300, and a matrix size of 32 ×4 reconstructed to 64 ×8). **f** Corresponding anatomical, fat-only, $^1$H image acquired on the same participant using the dual-tuned $^1$H/$^{129}$Xe coil. **g** Overlay of the HPXe image shown in (**e**) onto the corresponding anatomical fat-only $^1$H image shown in (**f**). **h** Coronal view of the $^{18}$F-FDG maximum-intensity-projection (MIP) image. Dotted red line indicates the location of the corresponding fused image shown in (**i**). **i** Overlay of the axial $^{18}$F-FDG map onto the corresponding anatomical fat-only axial $^1$H image showing increased glucose uptake in supraclavicular fat, where most of the $^{129}$Xe signal was observed.

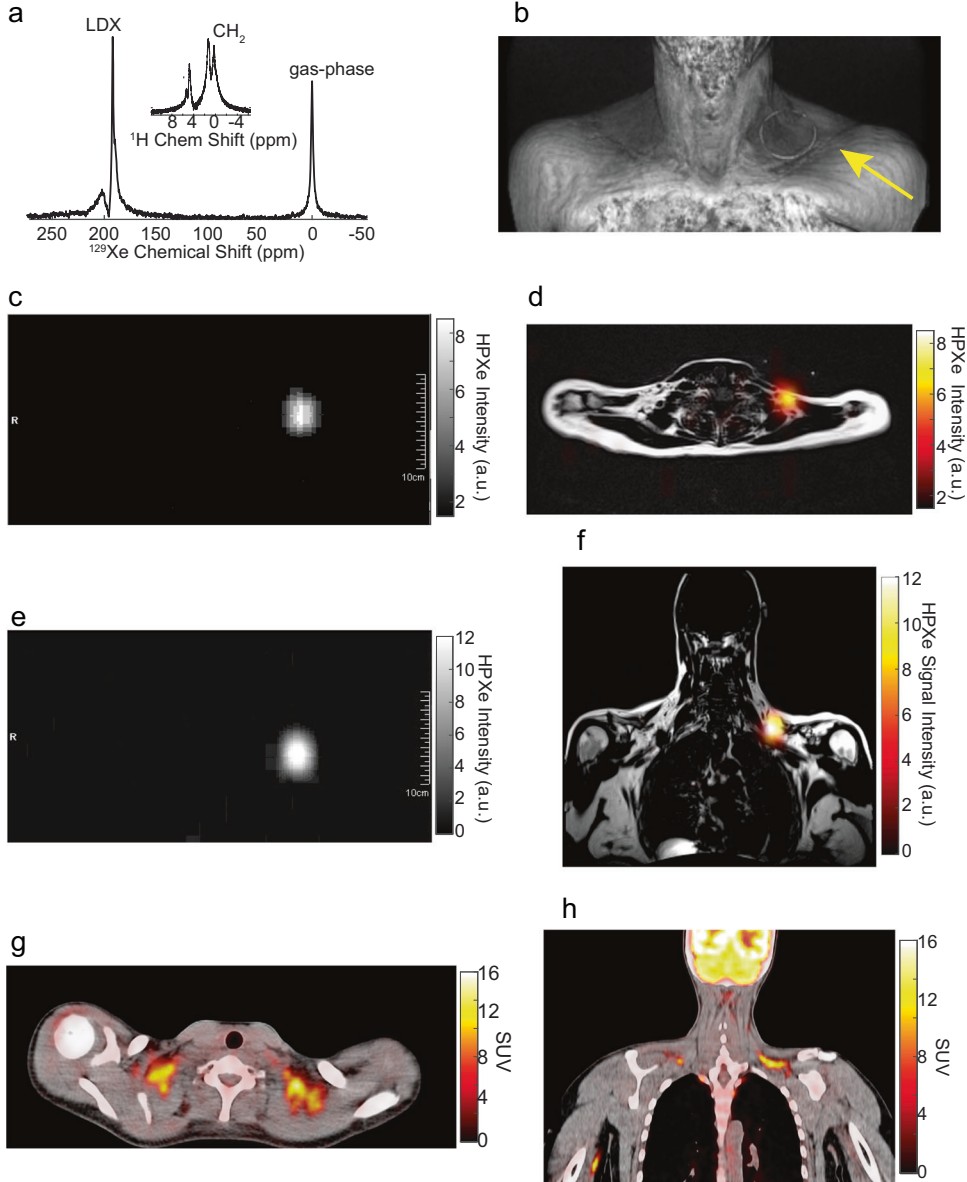

**Fig. 6 Enhanced xenon uptake in BAT detected by MRI. a** $^{129}$Xe and $^1$H (inset) spectra acquired during cold exposure on a 25-30 y.o. female with a BMI of 27 kg/m$^2$, showing a large narrow lipid-dissolved xenon (LDX) signal (linewidth 0.7 ppm) and the corresponding CH$_2$ peak, which appears split in two peaks. The distortion of the CH$_2$ peak led to a large temperature uncertainty of more than 2.5 °C ((37.5 ± 2.4) °C). In this case, reference to one peak would give a temperature value of 39.9 °C and reference to the other would give a temperature value of 35 °C. **b** 3D MRI volume rendering showing the location of the surface coil placed right above the supraclavicular fossa. A circular plastic tube containing water doped with CuSO$_4$ was placed on the surface of the $^{129}$Xe coil to localize its position relative to the supraclavicular fossa. **c** Axial $^{129}$Xe image (FOV = 225 ×450, MTX 8 × 32, TR = 25 ms, TE = 2.1 ms, NA = 12 BW = 430 Hz/Pixel) as reconstructed online from the scanner. **d** Overlay of the axial $^{129}$Xe image shown in (c) onto the corresponding anatomical $^1$H, fat-only, image showing localized $^{129}$Xe uptake in BAT. **e** Coronal $^{129}$Xe image (FOV = 225 × 450, MTX 8 ×32, TR = 25 ms, TE = 2.1 ms, NA = 12, BW = 430 Hz/Pixel) as reconstructed by the MR scanner. **f** Overlay of the coronal $^{129}$Xe image shown in (**e**) onto the corresponding anatomical fat-only $^1$H image. **g** Fused PET/CT axial image obtained from the same person showing increased glucose uptake in the supraclavicular region. **h** Fused PET/CT coronal image obtained from the same participant showing increased glucose uptake in the supraclavicular fat depot.

using the larger $^{129}$Xe surface coil and its intensity did not increase between thermoneutrality and cold exposure, suggesting it originated from regions outside of BAT. A broad small peak was also observed in some participants when using the larger $^{129}$Xe surface coil. This peak, located at 217 ppm from the gas phase peak and at about 25 ppm downfield from the LDX peak, originates from $^{129}$Xe spins bound to hemoglobin in red blood cells and most likely arose from major blood vessels located in the sensitive region of the larger surface coil.

Figure 5 and Supplementary Fig. 2 show some of the first $^{129}$Xe images acquired in this study by using a dual tuned $^1$H/$^{129}$Xe surface coil. These images show that, during cold exposure, the LDX signals observed in $^{129}$Xe spectra originate from the glucose-avid supraclavicular fat depot, where most of the BAT is in humans. Figures 6–8 and Supplementary Fig. 3-4 show $^{129}$Xe spectra and images acquired on healthy participants later in the study by using the single-tuned $^{129}$Xe surface coils on the PRISMA scanner. In this case, the use of a much smaller

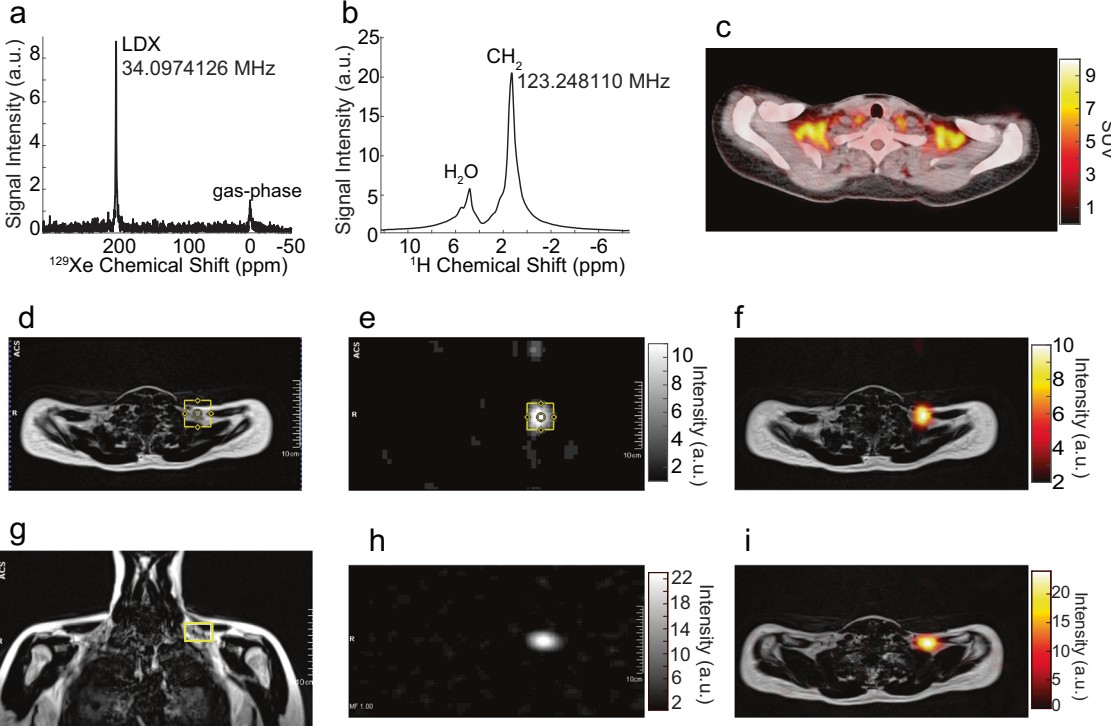

**Fig. 7 Detection of BAT thermogenic activity by absolute MR thermometry with hyperpolarized xenon. a** Hyperpolarized xenon (HPXe) NMR spectrum acquired during cold exposure at 8 s from the beginning of the breath hold, on a 25-30 y.o. female with a BMI of 24.1 kg/m$^2$. **b** Localized $^1$H spectrum acquired right before the acquisition of the HPXe spectrum shown in (**a**) by using a single-voxel spectroscopy sequence from the 40 ×40 x 26 mm$^3$ voxel centered within the supraclavicular fat depot and shown in yellow in (**d**), (**e**), and (**g**). The lipid-dissolved xenon (LDX) frequency, referenced to the $CH_2$ frequency, revealed active BAT with an average temperature of 39.6 ± 0.8 °C. **c** Fused axial $^{18}$F-FDG-PET/CT image collected from the same participant two days prior to the collection of the MR images and spectra, showing enhanced glucose uptake in the supraclavicular fat depot. **d** Anatomical fat-only MR axial image, showing the location of the voxel from which the spectrum in (**b**) was acquired. **e** Low-resolution HPXe MR image collected from the same participant showing enhanced $^{129}$Xe signal from the same shimmed region from which the $CH_2$ signal was acquired for temperature measurements.
**f** Overlay of the HPXe image shown in (**e**) onto the corresponding anatomical fat-only image. **g** Coronal fat-only $^1$H MR image acquired from the same participant showing the other dimension of the selected voxel from which the spectrum in (**b**) was acquired; **h** Second axial HPXe image acquired on the same participant on a different day. **i** Overlay of the HPXe MR image shown in figure (**h**) onto the corresponding anatomical fat-only $^1$H image. Enhanced xenon signal is observed from the same region within the supraclavicular fossa.

surface coil tuned to the lipid-dissolved $^{129}$Xe resonance enabled the acquisition of $^{129}$Xe images and spectra with a much higher resolution and SNR, which increased from an average of 3 ± 2 to an average of 12 ± 4.

**Measurement of BAT absolute temperature in humans during cold exposure.** BAT absolute temperature measurements were performed in 14 of the 18 participants analyzed during cold exposure, when increased tissue perfusion led to rapid uptake of $^{129}$Xe in BAT and to a large LDX peak. Figure 6–8 and Supplementary Fig. 3 and 4 show examples of the acquired spectra used to determine the LDX and $CH_2$ frequencies needed for absolute temperature measurements, while Table 1 reports all temperature measurements obtained under this study, when $^1$H and $^{129}$Xe spectra were acquired back-to-back using the same shim setting during the breath hold (Supplementary Data 3). The table also includes data from the participants in which spectral lines were broad or distorted, leading to large temperature uncertainties. An example of these spectra is shown in Supplementary Fig. 3 and 5. As can be seen from Table 1, in almost all participants we measured a temperature higher than 37 °C. A two-sided two-sample t-test indicates that the average BAT temperature measured in our participant during cold exposure was significantly (p < 0.02) higher than 37.7 °C, which is regarded as the upper

limit of the normal temperature range in healthy adults aged 40 years or younger[45]. Not surprisingly, a BAT temperature higher than 37 °C was observed also in two of the three participants that had a very hydrated supraclavicular BAT but did not show an increase in glucose uptake in their BAT (Fig. 8 and Supplementary Fig. 4).

**Discussion**
To the best of our knowledge, this study demonstrates the first direct detection of human BAT and thermogenic activity by MR with laser-polarized $^{129}$Xe. In vivo human studies showed a detectable lipid-dissolved xenon resonance that consistently increased during cold exposure in all participants, including those that showed a negative PET scan. Imaging experiments confirmed that this resonance originated primarily from $^{129}$Xe spins located within the supraclavicular fat depot, where BAT is known to be present in humans. Overall, these results mimic the results previously obtained in rodents and non-human primates, which have shown a selective increase in xenon uptake in the brown fat of anesthetized mice and non-human primates during stimulation of thermogenesis by catecholamines[34,38,42]. In humans, the absence of a strong lipid peak before stimulation of thermogenesis confirms that at thermoneutrality BAT is not very well perfused, leading to reduced wash-in rate and low uptake of polarized $^{129}$Xe atoms in the tissue. During cold exposure, the specific increase in

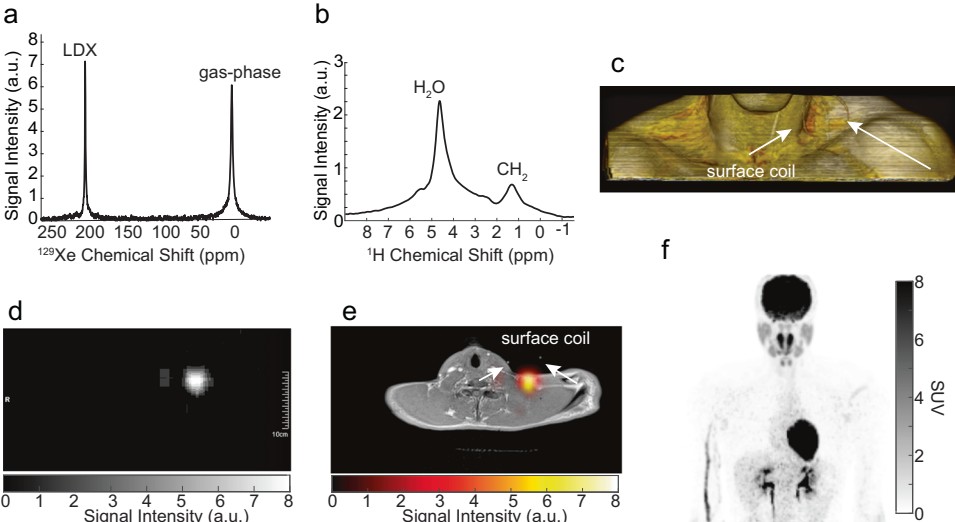

**Fig. 8 Hyperpolarized $^{129}$Xe spectrum and image acquired on a 25-30 y.o. male with a BMI of 29.9 kg/m$^2$ that showed enhanced xenon uptake in non-glucose avid supraclavicular fat.** The $^{18}$F-FDG-PET/CT was performed in the winter months with an outside temperature averaging 4 °C. **a** Hyperpolarized xenon (HPXe) spectrum acquired during cold exposure, using the small, single-tuned, xenon surface coil. The $^{129}$Xe NMR spectrum shows a large lipid-dissolved xenon peak at 193.2 ppm downfield from the gas phase peak. **b** Localized $^1$H NMR spectrum, acquired from a 30 mm × 30 mm × 30 mm voxel centered within the supraclavicular fat pad, showing a small CH$_2$ peak at 1.3 ppm. The CH$_2$ peak is much smaller than the water peak, indicating a very hydrated BAT. The lipid-dissolved xenon frequency, referenced to the CH$_2$ frequency, revealed active BAT during cold exposure with a temperature of (39.5 ± 2.5) °C. **c** 3D MR volume rendering of the same participant showing the positioning of the small surface coil labeled with a circular tube containing CuSO$_4$-doped water; **d** HPXe axial MR image showing localized $^{129}$Xe signal. **e** Overlay of the axial HPXe MR image shown in (**d**) onto the corresponding $^1$H image confirming that the lipid-dissolved $^{129}$Xe signal originates primarily from the supraclavicular fat pad. **f** $^{18}$F-FDG-PET maximum intensity projection-image, showing absence of glucose avid BAT in the supraclavicular region or other well-known BAT locations (axillary, paravertebral).

blood flow to BAT[3] leads to an increase of the xenon wash-in rate and selective uptake in BAT.

Along with the selective uptake of xenon in BAT observed during cold exposure in humans, we also report the first direct measurement of its temperature by absolute $^{129}$Xe MR thermometry. This methodology relies on the high solubility of xenon in lipids, on the strong temperature dependence of its chemical shift, and on the ability to use the temperature-insensitive resonance frequency of nearby lipid protons (CH$_2$ methylene protons) as a local reference to make the measurement independent of field strength and shimming conditions. Our in vitro studies confirmed the previously found linear temperature dependence of the chemical shift of lipid-dissolved $^{129}$Xe in the 25-40 °C temperature range. Additionally, these studies show that by using the nearby lipid protons as an internal temperature insensitive resonance, one can remove the effect of field drift, variation in shimming conditions, and magnetic field inhomogeneities, and extract absolute tissue temperature information. In vivo studies in rodents confirmed that the calibration performed in vitro of this NMR-based thermometer could be used to correctly determine adipose tissue temperature with an accuracy better than 0.2 °C. In this case, measurements were obtained with an average uncertainty of 0.5 °C.

As the increase in blood flow to BAT is a necessary but not sufficient condition for the increase in thermogenic activity[46], measurements of BAT temperature represent the most direct and accurate way to detect the thermogenic activity of this tissue. In our participants, direct temperature measurements of BAT were possible whenever $^1$H and $^{129}$Xe spectra were undistorted and sufficiently narrow that their position could be measured with enough accuracy (Supplementary Fig. 5). To achieve this goal, over the course of this study it appeared evident that HP $^{129}$Xe images, used to localize the region from which most of the LDX signal originated, and manual localized shimming procedures on the CH$_2$ protons from the same region that displayed enhanced

lipid-dissolved signal were both necessary, as was the use of the smaller xenon surface coil to spatially localize the LDX signal. Additionally, to minimize temporal field variations between the two scans due to breathing motion, which in the supraclavicular region can be quite severe (Supplementary Fig. 6), $^1$H and $^{129}$Xe spectra had to be acquired back-to-back during a single breath hold and over the course of a few seconds. When this was done, temperature measurements with an uncertainty of less than 2 °C were consistently obtained with a success rate close to 100%.

It is important to note that the accuracy of the proposed absolute thermometry method is based on the spatial co-localization of the CH$_2$ and LDX spins. Given the high vessel density in BAT depots[47,48] and the high diffusivity of xenon in lipids[49], we do not expect $^{129}$Xe concentration gradients within BAT. Nonetheless, there may be contributions to the CH$_2$ signal from nearby white adipocytes, which are often found mixed with brown adipocytes in classical BAT regions[50]. However, because the CH$_2$ resonance frequency has a weak temperature dependence, temperature gradients between white and brown adipocytes within the supraclavicular fat depot are not expected to change the accuracy of the measurement.

Enhanced xenon uptake in supraclavicular BAT during cold exposure and a BAT temperature >37.7 °C was also observed in most but not all participants that displayed a BAT-negative PET scan. Lack of, or reduction in, BAT glucose uptake in the supraclavicular fat of healthy young lean persons, even in the presence of a strong increase in resting energy expenditure during cold exposure, has already been reported[51–53]. Our study suggests that, despite the lack of glucose-avid BAT, some of these individuals have healthy and functional BAT that did not undergo adaptive reductions, as previously hypothesized.

The present study has several limitations. First, participants were primarily young and healthy, with a body mass index (BMI) that encompassed the lean and the mildly overweight phenotype. A larger clinical trial in the obese and diabetic population should

**Table 1 Summary table of participants' characteristics, BAT temperature values, and glucose maximum standardized uptake value (SUVmax).**

| Participant Number | Sex | Age (years) | BMI (kg/m$^2$) | BAT SUVmax | Measured BAT Temperature (°C) |
|---|---|---|---|---|---|
| 1 | M | 20-25 | 25.1 | 19.9 | NA[a] |
| 2 | M | 20-25 | 22.1 | NA[b] | NA[a] |
| 3 | F | 25-30 | 23 | 4.4 | NA[a] |
| 4 | F | 20-25 | 20 | 0.6 | NA[a] |
| 5 | F | 25-30 | 22.2 | 30.1 | NA[a] |
| 6 | F | 25-30 | 27 | 10.1 | (37.5 ± 2.4,)[c] |
| 7 | M | 30-35 | 21.4 | 0.8 | (38.7 ± 0.6; 38.6 ± 0.7) (40 ± 0.9)[d] |
| 8 | F | 20-25 | 22.7 | 10.7 | (41.1 ± 3.4)[c] (41.5 ± 2.5)[c] (38.5 ± 1.5; 37.8 ± 2.5)[c] |
| 9 | M | 20-25 | 22 | 12 | (39.5 ± 0.8)[d] |
| 10 | M | 20-25 | 24.4 | 11.2 | 38.4 ± 0.8 |
| 11 | F | 25-30 | 17.7 | 10 | (35.9 ± 5.6)[c] |
| 12 | F | 25-30 | 19.5 | 8.6 | (39.1 ± 9.3; 38.8 ± 5.9)[c] |
| 13 | M | 25-30 | 29.9 | 0.6 | (42.6 ± 2.2[e]; 39.5 ± 2.5) |
| 14 | F | 20-25 | 24 | 0.7 | (37.9 ± 2.3) (37.1 ± 0.4) |
| 15 | F | 25-30 | 24.1 | 8.3 | (39.6 ± 0.8; 39.1 ± 1.1) |
| 16 | M | 25-30 | 26.5 | 0.8 | (40.2 ± 0.4; 38.8 ± 1.1) (40 ± 0.4; 40.1 ± 0.4) |
| 17 | M | 20-25 | 26.4 | 3.2 | (39.6 ± 0.4; 39.2 ± 0.4) |
| 18 | F | 25-30 | 19.6 | NA[b] | (39 ± 0.4) (38.7 ± 0.4) |

Parentheses are used to group temperature values obtained during the same imaging session, but no less than 20 min apart, using different bags of polarized gas and slightly different shimming conditions, while the participant continued to be exposed to cold temperatures.
[a]$^{129}$Xe spectroscopy studies were not performed for temperature measurements.
[b]FDG-PET studies were not performed.
[c]Distorted or split $CH_2$ peak.
[d]Small SNR due to low polarization.
[e]Participant moved during breath hold.

spectra acquired from the tissue during and after the breath hold, as previously done in the brain[56]. Here, by measuring the increase in LDX signal intensity between thermoneutrality and cold exposure, and by normalizing the LDX signal to the gas phase signal to account for differences in gas polarization, one can estimate the change in tissue perfusion. Based on the >2-fold increase in LDX signal observed between thermoneutrality and cold exposure in some of the early participants in which spectroscopy studies were conducted both at thermoneutrality and during cold exposure, one can estimate a >2-fold increase in BAT blood flow during cold exposure. This result is in qualitative agreement with the change in BAT perfusion previously observed with $^{15}$O-water PET in adult humans during cold exposure[3]. It is important to note that this is an underestimation of the actual increase in BAT blood flow. While at thermoneutrality, some of the LDX signal may originate from $^{129}$Xe atoms dissolved in the lipid compartment of nearby white adipose tissue cells, whereas during cold exposure the increase originates primarily from the increase in $^{129}$Xe uptake in the lipid droplets of BAT.

Although the HPXe MRI technique used here is currently only available in just a few research centers, the recent FDA approval of this technique for lung ventilation studies is expected to widen both its access as well as its clinical use. By virtue of being radiation free, this MR-based technique could potentially be used in healthy volunteers or children to assess changes in tissue perfusion and thermogenic activity over time or to better understand the effect of nutrition and diet on BAT stimulation, the latter of which, indeed, is still a matter of contention because of limited and inconsistent results previously reported[57,58].

### Data availability

The study protocol and statistical analysis are all described in the main manuscript. All data needed to evaluate the conclusions in the paper are present in the paper and in the Supplementary Data. The Supplementary_Data_1 file contains source data for Fig. 3. The Supplementary_Data_2 file contains source data for Fig. 4, and The Supplementary_Data_3 file contains source data for Table 1. Raw data related to this paper is securely stored on removable hard drives in our institution and may be requested from the corresponding author on reasonable request.

### Code availability

The software code used for processing spectroscopy data is made available online on GitHub[59], while the spectrum fitting procedure is publicly available online at the MATLAB Central File Exchange repository https://www.mathworks.com/matlabcentral/fileexchange/23611-peakfit-m.

be able to fully assess the sensitivity of this technique in the general human population. Second, in the participants studied, we measured only the average supraclavicular BAT temperature. Temperature mapping should also be possible by using chemical shift encoding techniques, especially when using higher $^{129}$Xe polarization values currently achieved by modern polarizers[54] and perhaps by using a multi-breath $^{129}$Xe inhalation protocol, recently used in humans to assess lung ventilation dynamics[55]. The latter, along with the relatively long relaxation time of $^{129}$Xe in lipids (Supplementary Fig. 7), should lead to enhanced accumulation of HPXe in BAT. However, because temporal field variations due to breathing motion in this area are quite severe, for these measurements one should consider breath-hold triggered and interleaved $^{129}$Xe and $^{1}$H acquisitions. Finally, while temperature measurements in BAT are a step toward direct measurement of BAT thermogenic activity, the final tissue temperature will depend not only on the ability of the tissue to produce heat, but also on tissue perfusion. Indeed, high tissue perfusion could potentially lead to lower BAT temperature values as heat is rapidly redistributed through the body by blood flow. We cannot exclude that this was indeed the case for participant 14, one of the three participants that resulted BAT-negative in PET scans and in which BAT temperature was measured to be less than 38 °C. Together, measurements of tissue thermogenic activity, coupled with measurements of tissue perfusion are expected to be able to differentiate thermogenically active from thermogenically inactive BAT. Although not done in this study, tissue perfusion could be directly derived from dynamic $^{129}$Xe

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

## Acknowledgements

The authors thank Jordan Jimenez, Soma Prum, and Kristine Baluyot for their help with the IRB protocol, participants recruitment, monitoring, and safety reporting. This work is supported by the National Institutes of Diabetes and Digestive and Kidney Diseases trough grant No. R01DK108231 and R01DK123206.

## Author contributions

R.T.B. conceptualized and supervised the study. L.K. contributed to the design of the study. L.H. polarized all gas for the in vivo experiments in rodents. L.Z., M.A., A.M., CM., and M.K. performed the MR experiments. A.B., M.A., A.M., M.K., and N.B. polarized the gas for all experiments. L.Z., M.A., and R.T.B. analyzed the experimental data. S.A. contributed to the analysis of the human data. R.T.B. wrote the manuscript and L.Z., M.A., A.B., and L.K. contributed to the editing of the manuscript. All authors read and approved the manuscript.

## Competing interests

The authors declare no competing interests.
