## [Peer Review File · Communications Medicine]

Reviewers' comments:

Reviewer #1 (Remarks to the Author):

In this paper the authors report on their work on using laser polarized xenon as a probe for non-invasive temperature measurements using NMR spectroscopy in brown adipose tissue (BAT). The chemical shift of xenon dissolved in fat (LDX) shows a relatively strong temperature dependence and the observed frequency for this resonance can as such be used as a probe for MR thermometry. The temperature-insensitive resonance frequency of hydrogen nuclei in methylene groups in the adipose tissue serves as a reference residing in the same physical compartment, which allows for the removal of frequency shifts in the LDX resonance caused by magnetic field inhomogeneities. In this way, the absolute temperature of the BAT can be measured. This interesting work builds on earlier work performed by the same research group, now adding evidence for the ability to apply the method in humans to detect the increased uptake of xenon in BAT after cold exposure and to measure temperature increase in BAT due to thermogenic activity. The experiments reported are sound and well-described. In addition, they show a logical build-up towards human subjects: 1) calibration of the temperature dependence of LDX in in vitro studies with fresh human adipose tissue samples, 2) validation experiments in mice with an optical temperature probe readings serving as a reference, 3) experiments in human subjects.

I would like to compliment the authors on their interesting work, described in a paper that reads very well. I do have some specific questions and comments.

Specific comments and questions:

1. P. 4, l.112-116: The authors state that their approach "should enable direct measurement of temperature changes in fatty tissues as well as absolute temperature.", and that "The latter can be obtained by referencing the resonance frequency of the LDX to that of nearby methylene protons (CH₂) originating from the same fat compartment in which xenon atoms diffuse(s!), such as to remove the effect of macro and microscopic magnetic susceptibility gradients, while retaining the temperature-sensitive information encoded in the LDX chemical shift." I would think that a prerequisite for this to work well is that the LDX is nicely and homogeneously distributed in the BAT. In that case, all xenon nuclei do have nearby methylene protons. The question is whether in vivo a homogeneous distribution may be expected, since there the xenon can only enter the BAT via the blood vessels, that are not necessarily homogeneously distributed over the tissue. If this is not the case, it might not be possible to correct for all microscopic field effects, which could influence the performance of the method. Can the authors reflect on this issue, preferably also in the discussion section?
2. P. 9, l.263: "WAT": please write "white adipose tissue" in full (this is the first occurrence).
3. P. 10, l.273: Do I understand correctly that in vivo temperature measurements were not always successful, likely due to insufficient shimming or spatio-temporal field variations? If so, please report the success rate. Also discuss what are the next steps towards a reliable tool for routine use (also see comment #5).
4. P.10, l.282: Check the reference to Fig. S7. I believe this should be to Fig. S8.
5. P.10, l.291: The Discussion section lacks clear conclusions. In addition, the limitations of the work presented have not been clearly described. Can the authors also reflect on the next steps forward? The work presented is an important next step (translation to humans), but what will come next on the way to application of the method for either medical studies or maybe even routine healthcare? In addition, please also reflect a bit further on the potential of such a tool. What scientific and/or clinical questions can hopefully be answered once it has become available? This deserves more attention, so as to better put the topic into perspective.

6. P.11, l.331: Although ethical approval for the human studies is discussed on page 12, I seem to miss the statement on compliance with ethical regulations related to the animal experiments in the manuscript (which the authors did check in the editorial policy checklist).
7. P.12, l.360: Please explain why 9 subjects underwent PET/MRI and 5 subjects underwent PET/CT.
8. P.14, l.430-431: Typo: "Enlangen" should be "Erlangen" (twice).
9. P.17, l.34: Ref. 34 seems to be incorrect. Do the authors mean MRM 2017 Sep;78(3):1070-107?
10. P.17, refs. 35 and 37 seem to be to the same source. Please check. Refs 40 and 45 are to the same paper as well. Please carefully check all references.
11. P.22, Fig.4. The lettering in the figure caption does not match the figure. In F: please show the inset more clearly. G: caption belongs to H and I is not mentioned in the caption. Please check.
12. P.22, l.600: Minor detail: the authors wrote: "LDX signal (...) fitted to the expected time-dependent signal behavior.". I think it should be the other way around. A fit function was fitted to the data points.
13. P.24, l.627: Figure 6E is showing a PET/CT image and not PET/MRI, as mentioned in the caption.
14. Supplementary Material, page 1: reference to Figure S7 is missing in the text. In addition, please explain why the T1 measurement experiments was performed.

Reviewer #2 (Remarks to the Author):

This paper describes studies of using hyper polarized xenon as a way to measure absolute temperatures in brown adipose tissue (BAT) in vivo using MRI. It builds on previous work from the same group, but this is the first study applying it to humans in vivo. A phantom calibration experiments, same as has been done before, and in vivo rodent study, similar to before, are described in addition to the human study. In the human studies 18F-FDG-PET scans are also performed. The use of hyper polarized xenon to measure temperature in BAT is an interesting approach and the current work is a good step towards showing its utility in the clinic, albeit for a rather specific application. I have some general comments as well as some more minor specific comments which should be clarified and addressed before publication can be recommended.

General comments

There seems to be quite a few acronyms used that are never defined or explained. Please go through and define and explain where needed, even if they're used in your earlier papers they should probably be re-defined and (briefly) explained in the paper as well.

The abstract could benefit from some more details. E.g., include (in vivo) accuracy and precision of the suggested method, and what field of view, spatial and temporal resolution is achieved, to get the readers an idea of this up-front. Also, mention the PET study which is an important part of the current in vivo human study in the abstract.

There are a lot of methods mentioned in the Result-section, but maybe this is according to journal guidelines as otherwise the results wouldn't make sense to readers reading the article sequentially. It does result in some things being covered twice though, so maybe just double check what the journal guideline states in this regard.

The figures are in general small, hard to understand and the captions don't describe all subfigures, especially for figures 4-7. In general, it would probably be better to reduce the number of subfigures

and make them larger instead. Also, label all the sub-figures with their own letter (as it is now you have up to 4 figures in a subfigure, which can be quite confusing), and make sure they're all mentioned in the captions. It seems like figures 5-7 show more or less the same thing in 3 different volunteers – it might be better to show larger figures from one (or two) cases and save the rest for the supplementary materials. In cases like Figure 6D, it seems like just the bottom right sub-figure includes all the information of the 4 sub-figures, so suggest reducing the number and making that one larger.

The spatial resolution of some of the scans are not immediately clear. Some figure captions mention "the voxel", when the figure seems to show something with higher resolution than a single voxel – e.g., Fig 6D. Does "MTX" have to do with the resolution? It's mentioned but never explained what it is.

Why are no (spatial) temperature maps of the in vivo data included? If you have the calibration you should be able to convert peak spacing to temperature and plot that.

Specific comments:

Line 43-44; "...in just a few..."

Line 52; cAMP=Cyclic adenosine monophosphate? Please define.

Line 54; ATP=adenosine triphosphate

Line 75; Suggest changing "...shim gradients and can only be..." to "...shim gradients but can only be..."

Line 83-83; Please add references for this statement.

Line 89-90; "but these methods require a pre-calibration of the temperature dependence in the tissue of interest which is hard to obtain in vivo" but isn't this the case for all methods? The PRF change of water was investigated in vivo and found to be ~0.008-0.012 ppm/°C in vivo depending on tissue type (see, e.g., review article by McDannold), the separation of the NAA peak and water peak was "calibrated" in vivo, this study is calibrated in vivo, etc. The only approach which, to this reviewer's knowledge, has not been calibrated in vivo is diffusion-based approaches where a few studies have assumed CSF and water have the same temperature dependence.

Line 99-100; "First, the chemical shift of ^{129}Xe dissolved in lipids (LDX) is several ppm away from the chemical shift of xenon dissolved in blood or muscle 29,30." For the benefits of the reader, could the authors briefly mention why this is.

Line 107-109; Can the authors add references for these two statements?

Line 139-140; Sub figure C seems to show all 4 samples?

Line 140-146; What's the benefit of calculating these reference values rather than just using ω_{CH2} and ω_{LDX} and find the peak separation between them?

Line 147; "uncertainty in ω_{LDX} and ω_{CH2} ," how was this uncertainty calculated/evaluated?

Line 150-151; Please add using to a and b. Maybe also clarify that these are for all 4 samples?

Line 165; Can you please (briefly) expand on what you mean with peak centroid, and how you determined it? Add a reference if available.

Line 165-166; What was the linewidth for the 1H? As you measure the shift for this peak, its width is also of interest.

Line 168; First mention of rLDX, please define?

Line 179; "Early" in what sense?

Line 210; To make the point of higher SNR, please calculate and state the SNR in the different cases.

Line 217-218; Fig 5-7 are three different volunteers?

Line 221; "temperature was significantly higher than 37°C." So, what was the temperature? Please include (spatial) temperature maps. Please include a table summarizing the results for all volunteers. How was this significance determined, and what was it (state p-value etc.).

Line 221-223; Please state what "sufficiently narrow" corresponds to in ppm? How did you know if it wasn't "sufficiently narrow"? A figure showing a case that worked and a case that didn't would be very beneficial for the reader, please include this. What are these distortions, and where did they come from? In how many cases did you get reliable measurements?

Line 244; Change "known to located" to "known to be located"

Line 307-308; Did you somehow make sure there were only BAT in the samples? If not, is there any risk different types of fat have different calibration curves re. the 129Xe shift compared to 1H?

Line 339; What's the normal body temperature for the mice? If the bore was kept between 35-40 °C, does it make sense that the mice in a few cases had a core body temperature below 35 °C according to figure 3c? And in quite a few cases above 40 °C (i.e., a fever?)?

Line 346-347; "...were performed on the CH2 resonance on a volume located right above the 129Xe surface coil, ..." in Fig 3 it looks like the coil was positioned on top of the animal, so unclear where the CH2 reference was? Was it an external reference?

Line 361-362; "...either the same day or less than a week apart..." suggest stating how many at each time point, or simply say they were all within one week.

Line 388; "...or a single-tuned 129Xe coil..." what coil was used for 1H imaging in this case?

Line 396; What does "MTX" mean?

Line 414-416; Would suggest moving this equation to the Methods section and not introduce an equation in the results section. Also, please insert the reference where this is previously described in detail.

Line 426; What was the time course, how long was the lowering of the temperature and how long was each 1°C increase kept before the next increase?

§434-435; "Anatomical 1H MR scans were run simultaneously with the PET acquisition" presumably

only on the PET-MR studies? What was done for the PET-CT where no MR anatomical scans could be acquired?

Figure 3; The insert in A is quite small, suggest making its own subfigure. Also. Please define what the red and green boxes are? FOV? What's the difference between the two inlays (why are the boxes different size)? Does the Signal Intensity between A and B correlate (i.e., is the signal for CH2 ~2x that of 129Xe?) In C, state if the error bars are +/- standard deviation, standard error, or ...? Over time, or space?

Figure 4; So, C is a zoomed in view of LDX from A and B? E and F are hard to understand. Is the signal (red areas) in E the same as in the inlay in F? Suggest adding a border to the inlay in F – it's hard to see what's what. How is F a volume rendering, looks like just a 2D map? There's no description of what H and I are? Why is it gray scale in H and color in I? Suggest showing in H where the slice shown in I is (suggest doing this for all figures showing orthogonal views like this). Could you also indicate approximate locations of the cooling pads?

Figure 5; C The volume rendering is fairly small, so it's hard to see the coil. Could you please make the coil a different color, e.g., red? D – what "other well-known BAT locations"? Only one spot seems to light up. Please add arrows pointing to what you refer to, and name the locations.

Figure 6; D it's hard to see any difference between the top and the bottom images to the left, and the caption doesn't seem to describe what they are. Why not just keep the bottom right of the 4 images in D, and maybe add the yellow box showing the voxel? That way it could be 4x as large and it would be much easier to see. Same with F – the left figure doesn't seem to show anything different than what's color-coded in the right figure?

Figure 7; What do the two bottom right subfigures in D add compared to the two left ones (with color overlay)?

Supplementary material

T1 measurement of Xenon – if the T1 depends on concentration, where you expect tens of micromolar in BAT and had hundreds of micromolar in the corn oil phantom), wouldn't an easy experiment be to scan the phantom at a few timepoints during the "few days" it took to reach the hundreds of micromolar concentration? This could give a better idea of what to expect in vivo, and seems like an easy enough experiment to perform?

Reviewer #3 (Remarks to the Author):

The authors present a novel and promising MRI technique to directly assess the temperature of human brown adipose tissue (BAT).

The manuscript is well written and the topic is relevant to the field and timely.

However, I have the following concerns:

- the data presented are from five individual subjects and they are presented in individual figures
- where is the data from the remaining ten individuals who participated in the clinical study
- It would be necessary to present some aggregate data (table or figure) with the results in the measurements from all individuals or from groups, e.g. BAT positive or BAT negative individuals.
- Furthermore, it would be great to compare the results of the temperature measurements to the results of the FDG-PET performed after cooling.
- A figure depicting the study flow would be really helpful for the reader to understand how the human

study was performed. I.e. to me it is not entirely clear at which time-point of the cold-exposure experiment the MRI temperature measurement was performed. Continuously during the cold exposure or directly after cold exposure?

Discussion:

Quantification of BAT activity by direct measurement of tissue temperature represents a big opportunity to circumvent surrogate parameters such as FDG uptake. However, the authors should discuss that increased perfusion of BAT which follows its activation will reduce the temperature and needs to be an integral part of the assessment of BAT activity.

Minor:

There is a small typo in the location of Siemens Healthineers. It is "Erlangen"

We would like to thank all three reviewers for providing constructive feedback that helped improve the readability and quality of our manuscript. The manuscript has been substantially revised and additional data (including additional human data) has been added to respond to all reviewers critiques. Below is a point-by-point response to all remarks that have been made.

Reviewer #1:

In this paper the authors report on their work on using laser polarized xenon as a probe for non-invasive temperature measurements using NMR spectroscopy in brown adipose tissue (BAT). The chemical shift of xenon dissolved in fat (LDX) shows a relatively strong temperature dependence and the observed frequency for this resonance can as such be used as a probe for MR thermometry. The temperature-insensitive resonance frequency of hydrogen nuclei in methylene groups in the adipose tissue serves as a reference residing in the same physical compartment, which allows for the removal of frequency shifts in the LDX resonance caused by magnetic field inhomogeneities. In this way, the absolute temperature of the BAT can be measured. This interesting work builds on earlier work performed by the same research group, now adding evidence for the ability to apply the method in humans to detect the increased uptake of xenon in BAT after cold exposure and to measure temperature increase in BAT due to thermogenic activity. The experiments reported are sound and well-described. In addition, they show a logical build-up towards human subjects: 1) calibration of the temperature dependence of LDX in in vitro studies with fresh human adipose tissue samples, 2) validation experiments in mice with an optical temperature probe readings serving as a reference, 3) experiments in human subjects.

I would like to compliment the authors on their interesting work, described in a paper that reads very well. I do have some specific questions and comments.

Specific comments and questions:

1. P. 4, 1.112-116: The authors state that their approach “should enable direct measurement of temperature changes in fatty tissues as well as absolute temperature.”, and that “The latter can be obtained by referencing the resonance frequency of the LDX to that of nearby methylene protons (CH₂) originating from the same fat compartment in which xenon atoms diffuse(s!), such as to remove the effect of macro and microscopic magnetic susceptibility gradients, while retaining the temperature-sensitive information encoded in the LDX chemical shift.” I would think that a prerequisite for this to work well is that the LDX is nicely and homogeneously distributed in the BAT. In that case, all xenon nuclei do have nearby methylene protons. The question is whether in vivo a homogeneous distribution may be expected, since there the xenon can only enter the BAT via the blood vessels, that are not necessarily homogeneously distributed over the tissue. If this is not the case, it might not be possible to correct for all microscopic field effects, which could influence the performance of the method. Can the authors reflect on this issue, preferably also in the discussion section?

As this reviewer pointed out, the accuracy of the temperature measurement depends on the spatial co-localization of CH₂ and ¹²⁹Xe spins. Brown adipose tissue is a highly vascularized tissue, with each adipocyte being fed by at least one nearby capillary that directly transports O₂ and nutrients to the cell. Below is a cartoon of the difference in perfusion between brown and white adipocytes. The high vascularization and the specific increase in tissue perfusion that occurs during stimulation of non-shivering thermogenesis is responsible for the selective uptake of xenon in BAT during the first second of inhalation.

Image taken from : Lim, S., Honek, J. & Cao, Y. Blood Vessels in White and Brown Adipose Tissues. in *Angiogenesis in Adipose Tissue* (ed. Cao, Y.) 77–102 (Springer New York, 2013).
doi:10.1007/978-1-4614-8069-3_5.

Its high vessel density guarantees that ^{129}Xe is directly transported to each brown adipocyte, while the high diffusivity of xenon in lipids guarantees that it homogeneously distributes within its lipid compartment within few milliseconds. With that being said, there may be contributions to the CH_2 signal from nearby white adipocytes, which are often found in classical BAT regions, especially in humans.

These are not as vascularized as brown adipocytes and their mitochondria lack the UCP1 protein used by BAT for heat production. However, because the CH_2 resonance frequency has a weak temperature dependence, temperature gradients between white and brown adipocytes are not expected to change the accuracy of the measurement, unless the CH_2 frequency distribution in the region is very heterogeneous. This point is now made in the discussion.

2. P. 9, l.263: “WAT”: please write “white adipose tissue” in full (this is the first occurrence).

This has been spelled out

3. P. 10, l.273: Do I understand correctly that in vivo temperature measurements were not always successful, likely due to insufficient shimming or spatio-temporal field variations? If so, please report the success rate. Also discuss what are the next steps towards a reliable tool for routine use (also see comment #5).

That is correct. The magnetic field distribution in the supraclavicular region is extremely sensitive to breathing motion as shown in Figure S6 of the supplementary material which, along with the re-setting of gradient currents between ^1H and ^{129}Xe scans, made spectroscopy studies collected earlier unusable. Later in the study, these temporal field variations were minimized by consecutively acquiring ^1H and ^{129}Xe spectra back-to-back during the breath hold, within a couple of seconds, while shim currents were manually copied from the ^1H to the ^{129}Xe scans, before acquisition. When collecting data this way, the only issue we encountered was suboptimal shimming of the CH_2 peak. In some cases, this peak appeared split, suggesting that this signal originated from two different compartments, possibly from BAT and from nearby fat regions. Later this effect was further reduced by shimming a much larger area and then collecting the ^{129}Xe signal from a much smaller voxel from which most of the ^{129}Xe signal arose from. We are now reporting all temperature measurements performed in this study on table 1 in the order they

were acquired- including the initial measurements that were less successful because of poor shimming conditions.

4. P.10, l.282: Check the reference to Fig. S7. I believe this should be to Fig. S8.

We have corrected this reference.

5. P.10, l.291: The Discussion section lacks clear conclusions. In addition, the limitations of the work presented have not been clearly described. Can the authors also reflect on the next steps forward? The work presented is an important next step (translation to humans), but what will come next on the way to application of the method for either medical studies or maybe even routine healthcare? In addition, please also reflect a bit further on the potential of such a tool. What scientific and/or clinical questions can hopefully be answered once it has become available? This deserves more attention, so as to better put the topic into perspective.

Following this reviewer suggestion, we have added a discussion about the limitations of this study and have added a paragraph describing the potential of the proposed methodology for answering outstanding questions in the field. We believe that the observation of active BAT in subjects that had a negative PET-scan already demonstrated that in these subjects BAT is still very active, despite lack of glucose uptake.

6. P.11, l.331: Although ethical approval for the human studies is discussed on page 12, I seem to miss the statement on compliance with ethical regulations related to the animal experiments in the manuscript (which the authors did check in the editorial policy checklist).

The statement has now been added.

7. P.12, l.360: Please explain why 9 subjects underwent PET/MRI and 5 subjects underwent PET/CT.

The choice was made mostly based on PET/MRI and PET/CT availability and subject availability (please note that the PET scan involved significant time commitment from the subject considering the lengthy cooling procedure that had to be performed before the actual scan) within the same week when the xenon MR scan was performed.

8. P.14, l.430-431: Typo: “Enlangen” should be “Erlangen” (twice).

This typo has been corrected.

9. P.17, l.34: Ref. 34 seems to be incorrect. Do the authors mean MRM 2017 Sep;78(3):1070-107?

This reference has been corrected.

10. P.17, refs. 35 and 37 seem to be to the same source. Please check. Refs 40 and 45 are to the same paper as well. Please carefully check all references.

Thank you for noticing the issue with the references. These have all been checked for correctness and duplications.

11. P.22, Fig.4. The lettering in the figure caption does not match the figure. In F: please show the inset more clearly. G: caption belongs to H and I is not mentioned in the caption. Please check.

This figure has been revised accordingly.

12. P.22, l.600: Minor detail: the authors wrote: “LDX signal (...) fitted to the expected time-dependent signal behavior.”. I think it should be the other way around. A fit function was fitted to the data points.

We have rephrased the caption.

13. P.24, l.627: Figure 6E is showing a PET/CT image and not PET/MRI, as mentioned in the caption.

This indeed was a CT image. The caption has been corrected.

14. Supplementary Material, page 1: reference to Figure S7 is missing in the text. In addition, please explain why the T1 measurement experiments was performed.

Along with tissue perfusion and xenon solubility, the T_1 of xenon in the tissue of interest is one of the most important parameters that one must consider when imaging the dissolved phase. This is because a long T_1 enables signal build up and imaging past xenon exhalation- further reducing contamination from the gas phase (which here was pushed to the edge of the FOV using an appropriate bandwidth/voxel). A long T_1 may also be advantageous when using a continuous HPXe inhalation protocol, which has already been used for human lung imaging and could potentially be used here to improve image resolution. Because our in vivo dynamic experiments suggested a relatively long T_1 for xenon dissolved in lipids, we thought that these measurements would be helpful to include in this manuscript. This point is now clarified in the supplementary material and in the discussion.

Reviewer #2 (Remarks to the Author):

This paper describes studies of using hyper polarized xenon as a way to measure absolute temperatures in brown adipose tissue (BAT) in vivo using MRI. It builds on previous work from the same group, but this is the first study applying it to humans in vivo. A phantom calibration experiments, same as has been done before, and in vivo rodent study, similar to before, are described in addition to the human study. In the human studies ^{18}F -FDG-PET scans are also performed. The use of hyper polarized xenon to measure temperature in BAT is an interesting approach and the current work is a good step towards showing its utility in the clinic, albeit for a rather specific application. I have some general comments as well as some more minor specific comments which should be clarified and addressed before publication can be recommended.

General comments

1. There seems to be quite a few acronyms used that are never defined or explained. Please go through and define and explain where needed, even if they're used in your earlier papers they should probably be re-defined and (briefly) explained in the paper as well.

We have removed some of the acronyms that were introduced, especially when they were used only once or twice. We have also defined those that had been previously introduced.

2. The abstract could benefit from some more details. E.g., include (in vivo) accuracy and precision of the suggested method, and what field of view, spatial and temporal resolution is achieved, to get the readers an idea of this up-front. Also, mention the PET study which is an important part of the current in vivo human study in the abstract.

We have significantly revised the abstract to include important information including the experimental error with which these measurements were made, both in rodents and in humans, as well as the results of the validation study. Finally, we have also mentioned the use of ^{18}F FDG-PET, which is the most widely used technique to detect BAT activity.

3. There are a lot of methods mentioned in the Result-section, but maybe this is according to journal guidelines as otherwise the results wouldn't make sense to readers reading the article

sequentially. It does result in some things being covered twice though, so maybe just double check what the journal guideline states in this regard.

We have now conformed to the journal's style (please note that the manuscript had been transferred to Communication Medicine from another Nature journal) regarding manuscript outline.

4. The figures are in general small, hard to understand and the captions don't describe all subfigures, especially for figures 4-7. In general, it would probably be better to reduce the number of subfigures and make them larger instead. Also, label all the sub-figures with their own letter (as it is now you have up to 4 figures in a subfigure, which can be quite confusing), and make sure they're all mentioned in the captions. It seems like figures 5-7 show more or less the same thing in 3 different volunteers – it might be better to show larger figures from one (or two) cases and save the rest for the supplementary materials. In cases like Figure 6D, it seems like just the bottom right sub-figure includes all the information of the 4 sub-figures, so suggest reducing the number and making that one larger.

We have followed this suggestion and labeled each subfigure to clarify in the caption what each of them represent. We have also moved one of the figures in the supplementary material, while retaining one that shows HPXe maps of BAT in two orthogonal planes, one showing the repeatability of these maps (a subject that was scanned twice), and one showing enhanced glucose uptake and hot BAT in an FDG-negative subject. We have decided not to cut the number of subfigures to give the reader a better understanding of the region that was imaged (in one case only axially, in another both axially and coronally) as well as the quality of the HPXe maps. We would like to point out that high resolution images will be available through the journal's website, so the reader will be able to zoom in and out on the different parts of the figure.

5. The spatial resolution of some of the scans are not immediately clear. Some figure captions mention "the voxel", when the figure seems to show something with higher resolution than a single voxel – e.g., Fig 6D. Does "MTX" have to do with the resolution? It's mentioned but never explained what it is. Why are no (spatial) temperature maps of the in vivo data included? If you have the calibration you should be able to convert peak spacing to temperature and plot that.

We realized that this point was not clear in our manuscript. We have now specified that "voxel size" refers to the size of the voxel selected for single voxel ¹H spectroscopy experiments.

The HPXe images presented are collected by using a gradient recalled echo sequence, with a matrix size specified in the caption and presented as reconstructed by the clinical scanner. In other words, we did not do any post processing to the imaging data, except for the early one that could not be reconstructed directly by the scanner.

For temperature imaging (spatial temperature mapping) one would need to use a different pulse sequence to spatially encode the chemical shift, which was not used here. This point is also made in the discussion and also clarified in the abstract "global measurements of supraclavicular BAT temperature".

6. Specific comments:

Line 43-44; "...in just a few..."

We have changed this sentence as suggested.

Line 52; cAMP=Cyclic adenosine monophosphate? Please define.

Line 54; ATP=adenosine triphosphate

Both cAMP and ATP are now defined.

Line 75; Suggest changing "...shim gradients and can only be..." to "...shim gradients but can only be..."

We have made the suggested change.

Line 83-83; Please add references for this statement.

We have added references that show how, because water and fat spins generate from different water-fat compartments, they are inherently subjected to different local fields. Incidentally, this effect is also clearly visible in all human ^1H spectra we acquired: after we shimmed on the CH_2 peak, both the CH_2 and LDX peaks appear quite narrow, whereas the water peak becomes significantly distorted as it likely originated predominantly from nearby muscle tissue.

Line 89-90; “but these methods require a pre-calibration of the temperature dependence in the tissue of interest which is hard to obtain in vivo” but isn’t this the case for all methods? The PRF change of water was investigated in vivo and found to be $\sim 0.008\text{-}0.012$ ppm/ $^\circ\text{C}$ in vivo depending on tissue type (see, e.g., review article by McDannold), the separation of the NAA peak and water peak was “calibrated” in vivo, this study is calibrated in vivo, etc. The only approach which, to this reviewer’s knowledge, has not been calibrated in vivo is diffusion-based approaches where a few studies have assumed CSF and water have the same temperature dependence.

We realized that this sentence was not clear. What we meant is that the T_1 and its temperature dependence needs to be calibrated in the tissue of interest and at the field strength at which the measurement is done, whereas the PRF is relatively tissue and field independent. This sentence has been revised and references to support these statements have also been added.

Line 99-100; “First, the chemical shift of ^{129}Xe dissolved in lipids (LDX) is several ppm away from the chemical shift of xenon dissolved in blood or muscle 29,30.” For the benefits of the reader, could the authors briefly mention why this is.

The ^{129}Xe chemical shift is determined by the molecular composition of the tissue and the accessibility of the Xe to the various molecular groups. This explains why the chemical shift in lipids is significantly different than that in more aqueous environments like muscle or blood plasma. This is also now mentioned in the introduction.

Line 107-109; Can the authors add references for these two statements?

We have added references to previous work where xenon enhanced CT was used to quantify xenon concentration in different tissues (muscle-white fat and brown fat) at baseline and during stimulation of non-shivering thermogenesis. We have also added references to rodents, non-human primate, and human studies that have shown selective increase in BAT blood flow during stimulation of non-shivering thermogenesis.

Line 139-140; Sub figure C seems to show all 4 samples?

The figure shows the spectra acquired from only one of the samples but at the 4 different temperatures. This point is now clarified to avoid confusion.

Line 140-146; What’s the benefit of calculating these reference values rather than just using ω_{CH_2} and ω_{LDX} and find the peak separation between them?

The normalization by $\omega_{\text{CH}_2} \frac{\gamma_{\text{Xe}}}{\gamma_{\text{H}}}$ is used to obtain a field independent parameter, while the rescaling of ω_{CH_2} by $\frac{\gamma_{\text{Xe}}}{\gamma_{\text{H}}}$ is simply done out of convenience such that the temperature induced shift of ω_{rLDX} is the same as that of ω_{LDX} . This point is now clarified in the main manuscript.

Line 147; “uncertainty in ω_{LDX} and ω_{CH_2} ,” how was this uncertainty calculated/evaluated?

The uncertainty in the ω_{LDX} and ω_{CH_2} was extracted directly from the peakfit.m matlab routine, available at <https://www.mathworks.com/matlabcentral/fileexchange/23611-peakfit-m>

This fitting tool enables one to use bootstrapping to calculate the uncertainty in the estimation of peak position. Error propagation analysis was then used to obtain a temperature error out of the experimental uncertainty with which the two frequencies were determined and the uncertainty on the a and b coefficients that related the referenced LDX frequency to temperature. This point is now clarified in the manuscript.

Line 150-151; Please add using to a and b. Maybe also clarify that these are for all 4 samples?

We have edited this sentence as suggested to make it clearer.

Line 165; Can you please (briefly) expand on what you mean with peak centroid, and how you determined it? Add a reference if available.

As now mentioned in the manuscript, the fitting procedure was done by using the peakfit matlab routine publicly available online at <https://www.mathworks.com/matlabcentral/fileexchange/23611-peakfit-m>

Line 165-166; What was the linewidth for the 1H? As you measure the shift for this peak, its width is also of interest.

The linewidth of the CH₂ peak varied from subject to subject, but in most cases it ranged from 0.5 - 1 ppm. We are now reporting additional examples of spectra in the supplementary material that enable us to compare spectra that could not be used for temperature estimation to those that could.

Line 168; First mention of rLDX, please define?

rLDX is now defined when first introduced.

Line 179; “Early” in what sense?

We have changed the word early into “initial studies” to clarify that these were the first studies performed when we tried to use hyperpolarized ¹²⁹Xe to detect BAT in humans.

Line 210; To make the point of higher SNR, please calculate and state the SNR in the different cases.

The increase in SNR is now reported in the manuscript.

Line 217-218; Fig 5-7 are three different volunteers?

Yes, each figure shows data acquired in different volunteers.

Line 221; “temperature was significantly higher than 37°C.” So, what was the temperature? Please include (spatial) temperature maps. Please include a table summarizing the results for all volunteers. How was this significance determined, and what was it (state p-value etc.).

Following this reviewer suggestion, we have now included a table summarizing BAT temperature values measured in this study as well as SUVmax values measured in the supraclavicular fat pad of the volunteers studied. Please note that in the first 5 subjects temperature measurements could not be performed as shimming conditions were not the same for ¹H and ¹²⁹Xe spectral acquisitions. Additionally, the two spectroscopy scans were not collected back-to-back during a breath hold. We have also added two paragraphs describing how temperature errors were calculated, considering both the experimental uncertainty in peak position as well as uncertainty in the ω_{rLDX} time dependency found in vitro. A two-sample t-test was used to assess whether the average BAT temperature measured in this study was significantly different than 37.7°C, regarded as the upper limit of the normal oral temperature range in healthy adults aged 40 years or younger. These values are now included in the discussion.

Line 221-223; Please state what “sufficiently narrow” corresponds to in ppm? How did you know if it wasn’t “sufficiently narrow”? A figure showing a case that worked and a case that didn’t would

be very beneficial for the reader, please include this. What are these distortions, and where did they come from? In how many cases did you get reliable measurements?

We have included a figure in the supplementary material showing an example of sufficiently narrow spectra that enable temperature measurement, as well as those that did not. As now mentioned in the discussion, line distortion resulted from subject's movement during the breath hold, or from non-optimized shimming conditions (shimming conditions were optimized during free breathing, while spectra were acquired during a breath hold)

We have also added a table that includes all measurements made, both successful and unsuccessful measurements that contain large temperature errors because of the uncertainty in the peak position.

Line 244; Change “known to located” to “known to be located”

This typo has been corrected.

Line 307-308; Did you somehow make sure there were only BAT in the samples? If not, is there any risk different types of fat have different calibration curves re. the ^{129}Xe shift compared to ^1H ?

*No, we did not. However, as now specified in the introduction, the chemical shift difference between LDX and CH_2 protons and its temperature dependence is due exclusively to the chemical composition of the triglycerides molecules that make up most of the fat droplets in white and brown adipocytes. A detailed explanation of its molecular origin can be found in the discussion of: Antonacci et al, *Magn Reson Med*. 2019 Feb; 81(2): 765–772. In this manuscript it was shown that differences in adipose tissue composition, which across species are expected to be much larger than the differences between WAT and BAT in the same specie, do not produce appreciable changes in the LDX- CH_2 chemical shift and its temperature dependence.*

Line 339; What's the normal body temperature for the mice? If the bore was kept between 35-40 °C, does it make sense that the mice in a few cases had a core body temperature below 35 °C according to figure 3c? And in quite a few cases above 40 °C (i.e., a fever?)?

The plot includes all data points acquired for these studies. The data points below 35°C were the first data points acquired from one of the mice. Before the acquisition of this data point, the heating system had failed, and the mouse temperature had dropped. The data points above 41°C originated from a single mouse that we suspect entered a febrile state before dying. Nonetheless, these experiments also demonstrate the need for an independent and local temperature measurement, which in our case was possible by selecting tissue around the rectal probe.

Line 346-347; “...were performed on the CH_2 resonance on a volume located right above the ^{129}Xe surface coil, ...” in Fig 3 it looks like the coil was positioned on top of the animal, so unclear where the CH_2 reference was? Was it an external reference?

We have edited this figure as we realized that the orientation of one of the figures (taken as shown on the MR console) was misleading. The mouse was placed with its rectum on top of a small xenon surface coil. The mouse and the ^{129}Xe surface coil were then placed inside a large ^1H volume coil. The ^1H signal was collected from a voxel encompassing the sensitive region of the rectal probe and located right above the sensitive region of the ^{129}Xe surface coil.

Line 361-362; “...either the same day or less than a week apart...” suggest stating how many at each time point, or simply say they were all within one week.

We have changed the wording as suggested.

Line 388; “...or a single-tuned ^{129}Xe coil...” what coil was used for ^1H imaging in this case?

For all human studies we used the build-in ¹H body coil. This point is now clarified in the main text.

Line 396; What does “MTX” mean?

MTX stands from Matrix Size. We now defined this when it is first used.

Line 414-416; Would suggest moving this equation to the Methods section and not introduce an equation in the results section. Also, please insert the reference where this is previously described in detail.

This equation is now presented in the methods section, as suggested. The result section now contains the equation with the parameters obtained from the fitting of the in vitro data.

Line 426; What was the time course, how long was the lowering of the temperature and how long was each 1°C increase kept before the next increase?

For the PET study subjects were immediately exposed to cold temperatures right after changing into scrubs. The initial water was 18°C. Temperature was then lowered every 5 minutes, until subject started shivering. At that point we then increased the temperature by one degree every 5 minutes until the subject was no longer shivering. For the Xenon MR temperature study subjects were exposed to cold temperatures for about 20 minutes before temperature was performed. This point is now clarified in the manuscript.

\$434-435; “Anatomical 1H MR scans were run simultaneously with the PET acquisition” presumably only on the PET-MR studies? What was done for the PET-CT where no MR anatomical scans could be acquired?

For the PET-CT scans, CT images for attenuation correction were collected right before the acquisition of the PET data. This point has now been clarified.

Figure 3; The insert in A is quite small, suggest making its own subfigure. Also. Please define what the red and green boxes are? FOV? What’s the difference between the two inlays (why are the boxes different size)? Does the Signal Intensity between A and B correlate (i.e., is the signal for CH2 ~2x that of 129Xe)? In C, state if the error bars are +/- standard deviation, standard error, or ...? Over time, or space?

Figure 3 has been revised as suggested. The caption now explains what the different boxes represent, as well as how error measurement was calculated.

Figure 4; So, C is a zoomed in view of LDX from A and B? E and F are hard to understand. Is the signal (red areas) in E the same as in the inlay in F? Suggest adding a border to the inlay in F – it’s hard to see what’s what. How is F a volume rendering, looks like just a 2D map? There’s no description of what H and I are? Why is it gray scale in H and color in I? Suggest showing in H where the slice shown in I is (suggest doing this for all figures showing orthogonal views like this). Could you also indicate approximate locations of the cooling pads?

We have edited Figure 4 (now Figure 5) and its caption to clarify each part of this figure. The cooling pads are now shown in Figure S1. MIP images are generally shown using a linear black and white color-scale. The fused PET/MR image is automatically shown using a color scale by the DICOM reading software to enable differentiation between the anatomical MR image and the PET image overlaid on top. We have added a dotted line to show where the axial image shown in I is taken from.

Figure 5; C The volume rendering is fairly small, so it’s hard to see the coil. Could you please make the coil a different color, e.g., red? D – what “other well-known BAT locations”? Only one spot seems to light up. Please add arrows pointing to what you refer to, and name the locations.

Figure 5 (now Figure 6) has been edited. The volume rendering is now shown separately as the color cannot be changed. This figure is taken by reconstructing the MRI 3D data set. The coil is visible on the MR images as a circular water loop had been placed on its surface to identify its location with respect to the supraclavicular fat pad.

Figure 6; D it's hard to see any difference between the top and the bottom images to the left, and the caption doesn't seem to describe what they are. Why not just keep the bottom right of the 4 images in D, and maybe add the yellow box showing the voxel? That way it could be 4x as large and it would be much easier to see. Same with F – the left figure doesn't seem to show anything different than what's color-coded in the right figure?

This figure (now Figure 7) has been revised as suggested. Overlays generally give a better understanding of where the signal is originating from, while black and white maps give a better understanding of the SNR with which these maps are actually acquired. This is why we are keeping both.

Figure 7; What do the two bottom right subfigures in D add compared to the two left ones (with color overlay)?

Again, we think that by displaying the HPXe images alone, the reader can get a better sense of the overall signal SNR and potential contaminations that could be hard to appreciate on an overlay image. The noise, which is clearly visible in the two bottom right subfigures in D, cannot be seen in the overlay.

Supplementary material

T1 measurement of Xenon – if the T1 depends on concentration, where you expect tens of micromolar in BAT and had hundreds of micromolar in the corn oil phantom), wouldn't an easy experiment be to scan the phantom at a few timepoints during the “few days” it took to reach the hundreds of micromolar concentration? This could give a better idea of what to expect in vivo, and seems like an easy enough experiment to perform?

In solution, like in the gas phase, the reduction in T_1 is due primarily to Xe-Xe interactions. In solutions these interactions start to have an effect only at mM concentrations. At the small (micromolar) xenon concentrations expected in vivo in tissue after a single inhalation of the gas, these interactions are not expected to play a role. As such the T_1 measurement we have reported only provides a lower bound for the T_1 of xenon dissolved in lipids at 3T. This point is now clarified in the supplementary material. We would like to point out that we could not make measurements of LDX T_1 using physiological concentrations of xenon in tissue as the signal from thermally polarized ^{129}Xe spins at micromolar concentrations is too small to be detected by using our current 3T setup. The data shown were already collected over several hours.

Reviewer #3 (Remarks to the Author):

The authors present a novel and promising MRI technique to directly assess the temperature of human brown adipose tissue (BAT).

The manuscript is well written and the topic is relevant to the field and timely.

However, I have the following concerns:

- **the data presented are from five individual subjects and they are presented in individual figures**
- **where is the data from the remaining ten individuals who participated in the clinical study**
- **It would be necessary to present some aggregate data (table or figure) with the results in the measurements from all individuals or from groups, e.g. BAT positive or BAT negative individuals.**

As suggested, we have now included a table with all temperature values obtained from our subjects.

- **Furthermore, it would be great to compare the results of the temperature measurements to the results of the FDG-PET performed after cooling.**

In the added table we have also included SUVmax values for all subjects that underwent a PET scan.

- A figure depicting the study flow would be really helpful for the reader to understand how the human study was performed. I.e. to me it is not entirely clear at which time-point of the cold-exposure experiment the MRI temperature measurement was performed. Continuously during the cold exposure or directly after cold exposure?

As suggested, we have now added a figure showing the study flow. All HPXe temperature measurements had to be made during cold exposure. At thermoneutrality the LDX is close to noise level, and can be hardly detected. During cold exposure the signal dramatically increases and enable measurements of BAT temperature.

Discussion:

Quantification of BAT activity by direct measurement of tissue temperature represents a big opportunity to circumvent surrogate parameters such as FDG uptake. However, the authors should discuss that increased perfusion of BAT which follows its activation will reduce the temperature and needs to be an integral part of the assessment of BAT activity.

We agree with this statement. High tissue perfusion will enable quick dissipation of heat produced in BAT, therefore measurements of tissue perfusion should be coupled to measurements of the tissue's temperature. Incidentally, we believe that BAT activation cannot be ruled out even in the PET-negative subject that did not show a hot BAT. Most likely in this subject, that had a very hydrated BAT, we cannot rule out that blood flow was high enough to efficiently dissipate the heat produced in the tissue. Measurements of tissue blood flow could also be obtained by using dynamic hyperpolarized ¹²⁹Xe gas spectroscopy, after careful calibration of the other parameters that will contribute to the observed signal intensity (mostly careful calibration of the RF- which was not done in this study- transport of xenon in tissue and other relevant relaxation times). We have added a paragraph in the discussion about this issue.

Minor:

There is a small typo in the location of Siemens Healthineers. It is "Erlangen"

This typo has been corrected.

REVIEWERS' COMMENTS:

Reviewer #1 (Remarks to the Author):

I believe the authors have adequately addressed the comments made by the reviewers and that they have revised the manuscript accordingly. I have no further comments or questions.

Reviewer #2 (Remarks to the Author):

I would like to thank the authors for the detailed and thorough review of the manuscript. I think it is much clearer and the figures are better composed in the revised version. The authors have answered all my questions. My only (minor) suggestion is regarding Figure 1 - the ^{129}Xe atoms are depicted as gray in the top right part of the figure, but the ^{129}Xe spin is purple in the lower part of the figure (while the ^1H spin is now gray). Suggest keeping the color consistent within the figure.

Reviewer #3 (Remarks to the Author):

The authors amended the manuscript and answered my questions satisfactorily.

We were please to know that all three referees were satisfied with our revision. Below is our response to the last minor critique raised by Reviewer #2

Reviewer #2 (Remarks to the Author):

I would like to thank the authors for the detailed and thorough review of the manuscript. I think it is much clearer and the figures are better composed in the revised version. The authors have answered all my questions. My only (minor) suggestion is regarding Figure 1 - the ^{129}Xe atoms are depicted as gray in the top right part of the figure, but the ^{129}Xe spin is purple in the lower part of the figure (while the ^1H spin is now gray). Suggest keeping the color consistent within the figure.

Response: *We have edited Figure 1 and colored all ^{129}Xe atoms in purple, both in the upper and lower portion of the figure.*